# A biodegradable and flexible neural interface for transdermal optoelectronic modulation and regeneration of peripheral nerves

Pengcheng Sun[1,16], Chaochao Li[2,16], Can Yang[1,16], Mengchun Sun[2], Hanqing Hou[3], Yanjun Guan[2], Jinger Chen[1], Shangbin Liu[1], Kuntao Chen[1], Yuan Ma[4], Yunxiang Huang[5], Xiangling Li[2,6], Huachun Wang[7], Liu Wang[8,9], Shengfeng Chen[2], Haofeng Cheng[2], Wei Xiong[10], Xing Sheng ⓘ[11,12,13], Milin Zhang ⓘ[4], Jiang Peng[2,14], Shirong Wang ⓘ[15] ✉, Yu Wang ⓘ[2,14] ✉ & Lan Yin ⓘ[1] ✉

Optoelectronic neural interfaces can leverage the photovoltaic effect to convert light into electrical current, inducing charge redistribution and enabling nerve stimulation. This method offers a non-genetic and remote approach for neuromodulation. Developing biodegradable and efficient optoelectronic neural interfaces is important for achieving transdermal stimulation while minimizing infection risks associated with device retrieval, thereby maximizing therapeutic outcomes. We propose a biodegradable, flexible, and miniaturized silicon-based neural interface capable of transdermal optoelectronic stimulation for neural modulation and nerve regeneration. Enhancing the device interface with thin-film molybdenum significantly improves the efficacy of neural stimulation. Our study demonstrates successful activation of the sciatic nerve in rodents and the facial nerve in rabbits. Moreover, transdermal optoelectronic stimulation accelerates the functional recovery of injured facial nerves.

Implantable neural interfaces have been demonstrated for multiple therapeutic interventions[1–4] targeting various neurological abnormalities, such as Parkinson's syndrome[5,6], epilepsy[7,8], neuropathic pain[9,10], overactive bladder[11], depression[12,13], and nerve injuries[14,15], among others. Neural modulation based on direct electrical stimulation[16–18] and drug-based techniques[19,20] have been extensively studied. Conventional electrical stimulators have relatively large sizes and commonly necessitate connections to bulky batteries or external power supply equipment, potentially leading to undesired inflammations[21]. Recent endeavors in device miniaturization and integration with wireless circuits based on radio frequency (RF) and inductive coupling have shown promise in addressing these concerns[15,22,23]. Moreover, drug-based methods typically exhibit limited temporal resolution compared to electrical stimulations[24]. Other techniques, such as ultrasound[25–28], magnetoelectronics[29], and magnetism[30], have also

been explored to further achieve miniaturized and noninvasive or minimally invasive systems.

Optical interfaces represent one promising approach to achieving wireless neural stimulation without needing large-sized components. Given its high cellular specificity, optogenetics enables precise neural modulation and has been widely adopted in basic biomedical research[31–35]. However, the need for genetic modification constraints clinical translation. Alternatively, non-genetic optical neuromodulation has been proposed more recently, such as photothermal[36–39], optoelectronic[40–47] and photoacoustic systems[48,49]. Among these techniques, optoelectronic devices promise to achieve wireless and precise neural modulation without the need for excessive light intensity. These devices can convert light (red to near-infrared region 600-1000 nm) penetrated through tissues into electric currents and modulate neural activity via photocapacitive[41,42,50] and/or

photoelectrochemical[45,51] effects. While optoelectronic implants have demonstrated the potential for in vivo nerve stimulation[43,52,53] and ex vivo heart pacing[52], there remain significant hurdles to overcome in order to achieve clinically relevant applications and maximize therapeutic outcomes. This is because in the context of short-term stimulation (typically ranging from a few days to a few weeks) targeted at temporary treatments like tissue regeneration and pain management, the use of devices becomes unnecessary upon the completion of tissue repair or the mitigation of symptoms. Biodegradable systems can therefore offer the advantages of eliminating unnecessary material retention and avoiding the requirement for secondary surgery to remove the device, thereby maximizing therapeutic outcomes. Recent demonstrated examples include biodegradable electrical stimulators for sciatic nerve regeneration[15,54], pain management[55], etc.

As silicon (Si) has demonstrated desirable biocompatibility and biodegradability[56–59], devices based on Si membranes could offer potential solutions. However, interface decoration of Si devices with noble metals is often required to realize high power density, which would otherwise impede transdermal optoelectrical stimulation[45,50]. Electrical stimulation at the early stages of tissue injuries has been proven to effectively promote nerve regneration[15,54,60,61]. Electrical stimulation is believed to accelerate early Wallerian degeneration[62], stimulate calcium activity which upregulates regeneration associated genes (RAGs) through the cyclic adenosine monophosphate (cAMP) pathway[63], and increase the proliferation and production of neurotrophic factors of Schwann cells[60], promoting nerve regeneration. Effective transdermal optoelectronic stimulation could therefore potentially provide electrical cues wirelessly to facilitate nerve regrowth and functional restoration, which has been rarely explored. In all, optoelectronic devices that simultaneously satisfy full biodegradability and high stimulation efficacy for transdermal neural modulation and regeneration are yet to be developed. The biodegradable nature of the device can offer the unique advantage of eliminating retrieval procedures and minimizing associated infection risks.

Here, we propose a fully biodegradable, flexible and miniaturized optoelectronic device based on thin film Si diode and accomplish wireless transdermal stimulation and functional restoration of peripheral nerves (Fig. 1). The introduction of an interface modification layer of dissolvable molybdenum (Mo) enables highly efficient charge injection. Moreover, adopting extended electrodes allows a tripolar stimulation configuration to further assist stimulation[64]. Nerve activation is achieved by irradiating tissue-penetrating red light (635 nm) on the sciatic nerve of Sprague-Dawley (SD) rats and the facial nerve of New Zealand rabbit (Fig. 1). Successful functional recovery of injured facial nerves is achieved by transdermal optoelectronic stimulation. These results offer important avenues for developing of biodegradable, efficient, and miniaturized optoelectronic devices for transdermal neural modulation and associated regenerative medicine.

## Results
### Device structure and operational characteristics of the optoelectronic neural interface

One of the key challenges in achieving effective transdermal optoelectronic neural stimulation for optimal therapeutic outcomes is the development of biodegradable and efficient devices. While studies have suggested that metallic catalysts used to modify the surface of Si pn diodes can increase stimulation efficacy by facilitating capacitive and faradaic charge injections, with gold (Au) modification showing the most significant improvement[43,50], the development of fully biodegradable interface modification materials has yet to be explored for Si diodes. Here, we hypothesize that the introduction of a thin biodegradable metallic decoration layer based on Mo can potentially offer a feasible solution.

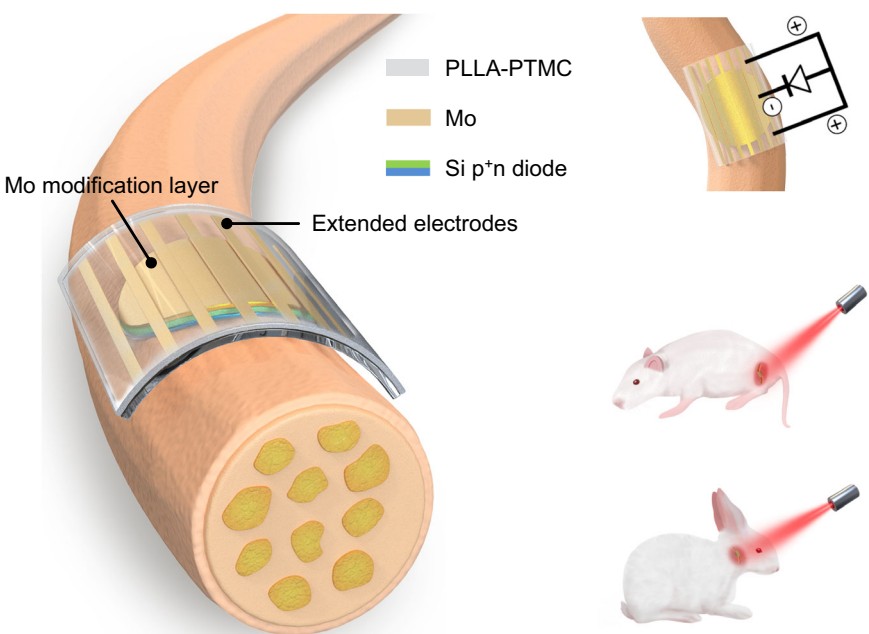

**Fig. 1 | Schematic illustration of the biodegradable and flexible neural interface for optoelectronic stimulation of peripheral nerves.** Optoelectronic neural interface consists of a thin film Si pn diode (p⁺n, 2.5 μm) with the n side in contact with nerve tissues, an interface modification layer of Mo (10 nm), and extended Mo electrodes (300 nm), on a flexible PLLA-PTMC substrate (200 μm). The biodegradable and flexible neural interface can achieve conformal contact with nerve tissues. Thin film Mo decoration at the device interface enables enhanced charge injection and improves neural stimulation efficiency. Equivalent circuit of the optoelectronic neural interface for nerve stimulation shows that extended electrodes enable a tripolar stimulation configuration that can further facilitate stimulation. By leveraging tissue penetrating red light (635 nm), the optoelectronic neural interface accomplishes effective stimulation of sciatic nerves on SD rats, and transdermal stimulation of facial nerves of New Zealand rabbits for functional recovery.

Legend (from figure): PLLA-PTMC, Mo, Si p⁺n diode. Labels: Mo modification layer, Extended electrodes.

The device consists of a thin film Si pn diodes (p⁺n, 2.5 μm), an interface modification layer (Mo, 10 nm) to achieve efficient charge injection, and an array of extended electrodes (Mo, 300 nm) to allow tripolar stimulation configuration (Fig. 1). The fabrication scheme is given in Supplementary Fig. 1. Specifically, thin film Si diodes (p⁺n, 2.5 μm) are made by p-doping of n-type silicon-on-insulator (SOI) wafers through ion-implantation. Thin films Si diodes are then patterned into miniaturized sizes (4 mm or 2 mm in diameter) by photolithography and reactive ion etching (RIE). The p⁺ side of Si diodes (p⁺n) is modified with a thin metallic layer (Mo or Au, 10 nm) by sputtering, to ensure high stimulation efficacy and sufficient light transparency. The scanning electron microscope (SEM) and energy-dispersive X-ray spectroscopy (EDS) results of the modified p⁺ surface appear in Supplementary Fig. 2. Si diode membranes are picked up by heat release tape and transfer-printed onto a biodegradable and soft polymeric substrate (copolymer of poly(L-lactic acid) and poly(trimethylene carbonate), PLLA-PTMC, 200 μm) with an array of extended Mo electrodes (300 nm), to enable a tripolar-electrode configuration which is expected to enhance local potential difference and facilitate stimulation[64]. Moreover, the PLLA-PTMC substrate possesses Young's modulus ($1.45 \pm 0.27$ MPa) similar to neural tissues[54], allowing minimal mechanical mismatch and conformal contact at the device/nerve interface. The n side of Si diodes (p⁺n) is placed in contact with neural tissues to perform stimulation, as n-type Si surfaces have successfully demonstrated light-evoked activation of peripheral nerves[53].

The photograph of the biodegradable Si-based optoelectronic interface appears in Fig. 2a. We use a previously reported patch-clamp set-up for photoresponse measurement[53], and the cross-sectional view of the device for measurement is given in Fig. 2b. The device is immersed in PBS, with the pipette positioned within 5 μm from the surface of the sample and a reference electrode in the PBS, mimicking in vivo conditions. As excellent photoresponse has been demonstrated by Si-based photovoltaic devices with deep red light[65], photocurrents and photovoltages are recorded under the illumination of a pulsed red laser beam (635 nm, 10 Hz, pulse width of 10 ms, spot size of ~ 2 mm, adjustable light intensity by controlling the input current to the laser diode (Supplementary Fig. 3). Photovoltages are generated within the Si diode via photovoltaic response by the separation of charge carriers (electrons and holes) upon illumination[66] (Fig. 2c). The photogenerated carriers can then migrate and accumulate at the Si surface and induce photocapacitive and/or photoelectrochemical effects. Specifically, on the n side of the Si diode (p⁺n), excessive electrons can lead to cell depolarization and stimulate action potentials of nerve tissues[53]. Holes on the p⁺ side are driven through the modification layer and extended electrodes which act as return electrodes to close the current path, enabling a tripolar electrode configuration (Fig. 1) that has been reported to improve stimulation efficacy[43,64].

The representative photovoltage responses of Si diode with Mo modification layer (Si/Mo device) under illumination (0.95 W/cm², safe for most tissues)[52,67] is shown in Fig. 2d. A cathodic voltage transient is observed upon light irradiation followed by an anodic transient when light is off, corresponding to the charging and discharging of the device, which imitates a biphasic stimulation mode that can modulate the behavior of adjacent nerve tissues. The measurement is found to be sensitive to the position and distance from the electrodes as well as the amount of liquid in the system, and the conditions are kept similar to ensure consistent results. In addition, to identify the influence of photothermal effects on photoresponse measurement, we evaluate the changes in temperature upon pulsed laser illumination of the device surface by an infrared thermal imaging camera, and the results suggest negligible increase in surface temperature after 5 s of irradiation (635 nm, 10 Hz, pulse width 10 ms, 0.95 W/cm²) on the device that immersed in PBS (Supplementary Fig. 4a). We also simulate the

transdermal condition by placing the Si/Mo device on the facial skin (~ 2 mm) of New Zealand rabbits with light incident from the back and passing through the skin. The result again indicates a minimal temperature increase of the device after 1 h of pulse laser illumination (Supplementary Fig. 4b).

To improve stimulation efficacy, the p⁺ side of the Si diode (p⁺n) is modified by a metallic layer. As the modification layer and extended electrodes are located on the back side of the device during in vivo applications, a thin modification layer (~ 10 nm) and a separated array of extended electrodes are employed to maintain desirable transparency, allowing light transmission to reach the Si diodes. To assess the effectiveness of the biodegradable modification layer, photoresponse is first evaluated with front-side illumination (Fig. 2b), excluding the influence that may be attributed to light transmission. Results with back-side illumination to demonstrate the potential effects of light absorption will be discussed later. We measure the photovoltage of Si diodes without modification layers (Si devices), Si diodes with a biodegradable Mo modification layer (Si/Mo devices), and Si diodes with a traditional Au modification layer (Si/Au devices), under the irradiation of continuous red pulsed laser beam with different light intensity (Fig. 2e). The results suggest a considerable enhancement in the photovoltage of Si/Mo and Si/Au devices compared to the devices without modification layers (Si devices). Mo modified device demonstrates the greatest photovoltage, which almost doubles the value of that of the Si/Au device. Moreover, the measured photovoltage increases with the increasing illumination intensity and reaches a plateau at light intensities greater than 0.95 W/cm². It is noted that the measured photovoltage (~ 30 mV) is lower than the open-circuit voltage of a conventional photovoltaic Si diode (~ 300-500 mV)[68], probably due to the presence of an additional device/solution interface. Nevertheless, photovoltage in the range of ~30 mV is expected to be sufficient for neural stimulation[53]. Figure 2f illustrates the corresponding measured photocurrents of Si, Si/Mo, and Si/Au devices upon laser irradiation (0.95 W/cm²) and the results demonstrate that Mo modified devices achieve the highest photocurrents, consistent with the measured photovoltages. In addition, the measured photoresponse demonstrates typical exponential transient decay, which is beneficial for efficient charge injection into the nerve and lowers the threshold for charge injection[69,70].

The total injected charges, and the contribution of the faradaic charge and the capacitive charge can be determined from the photocurrent transients under the illumination period (Supplementary Fig. 5) following a previously reported method[52]. The results are summarized in Figs. 2g, 2h and 2i, and Si/Mo devices demonstrate the highest total injected charges with a greatly enhanced faradaic charge and capacitive charge. Specifically, for the Si and Si/Au devices, the faradaic charge represents merely 1%-2% of the overall injected charge, whereas in Si/Mo devices, it accounts for up to 30%. The increased faradaic component of Si/Mo devices could be attributed to the potential electrochemical reaction of Mo with multiple oxidation states, especially in the presence of the positive photovoltage on the Mo modified p⁺ side of the Si diode. Together with enhanced interface capacitive components, the presence of Mo decoration can significantly improve charge injection efficacy. Detailed characteristics of the Mo modification layer will be discussed in the following session. Both Si/Au and Si/Mo devices exhibit desirable stability over 12000 cycles of pulse laser irradiation (635 nm, 0.95 W/cm², 10 Hz, pulse width 10 ms) (Supplementary Fig. 6a). The photocurrent of Si/Mo and Si/Au devices retain 94% ($I/I_0$) and 84% of the initial optoelectronic current ($I_0$) respectively, indicating a slightly lower retention current of the Si/Au device. This may arise from the adsorption of chloride ions on the surface of Au films[71], which could influence their microstructure. Additionally, potential microscale delamination could also contribute, as Au thin films are known to exhibit poor adhesion to Si surfaces[72,73].

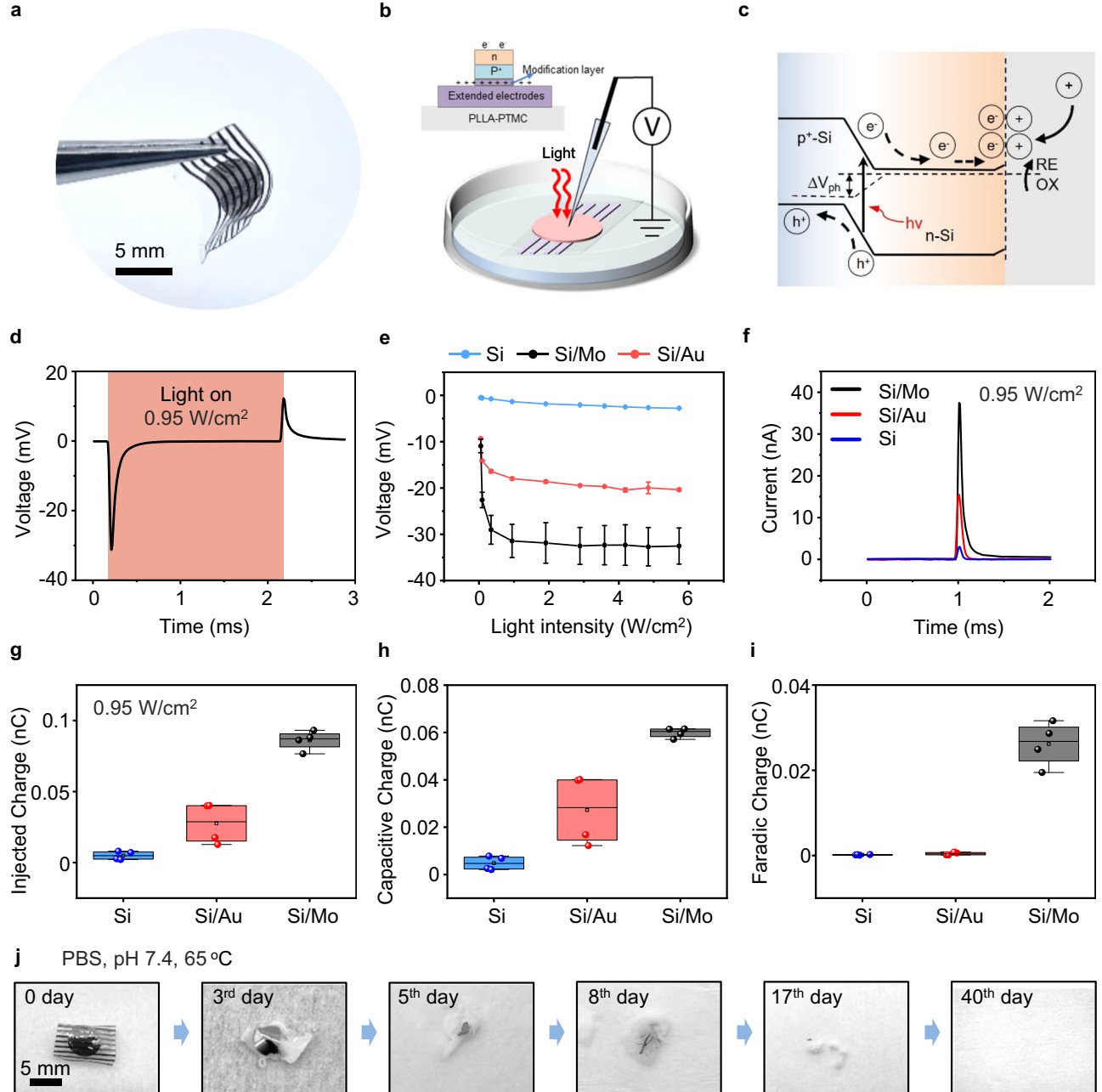

**Fig. 2 | Device structure and performance of biodegradable optoelectronic neural interfaces. a** Optical image of a biodegradable optoelectronic interface. **b** Schematic diagram of the patch-clamp set-up for the measurement of photoresponse. Light (635 nm laser, 10 Hz, pulse width of 10 ms, spot size of 2 mm) is incident from the front side of the device immersed in the PBS solution. The cross-sectional view of the device is given in the inset. **c** Schematic illustration of the photogenerated carriers within a pn junction, where illumination produces hole (h$^+$)-electron (e$^-$) pairs. The electrons can migrate to the Si/solution interface and induce photocapacitive and/or photoelectrochemical effects. Dashed line represents the quasi-Fermi level. **d** Representative photovoltage responses of the Si-based device with Mo modification layer (Si/Mo) upon illumination (0.95 W/cm$^2$). **e** Measured photovoltage of Si-based devices with different modification layers (Si/

Mo, Si/Au) and devices without interface modification layer (Si), at different light intensities. **f** Representative photocurrent responses of Si/Mo, Si/Au, and Si devices upon illumination (0.95 W/cm$^2$). **g, h** and **i** The total injected charge, capacitive charge, and faradaic charge calculated from measured photocurrent under illumination (0.95 W/cm$^2$) of Si/Mo, Si/Au, and Si devices. **j** Optical photographs of the degradation process of the biodegradable optoelectronic device in phosphate buffered saline (PBS, pH 7.4, 65 °C). In **a, j**, n = 3 samples. In **d, e, f**, n = 6 samples in each groups. In **g, h, i**, n = 4 samples in each groups. All data are presented as mean ± s.e.m. The box plot presents the median (center line), lower quartile (lower border), upper quartile (upper border), maximum (upper whisker) and minimum (lower whisker), which are ≤ 1.5 times the interquartile range.

For in vivo optoelectronic stimulation, light will be incident from the backside and transmitted through the semitransparent modification layer and reach the Si diode. The influence of light transmission is therefore investigated, and the results are given in Supplementary Fig. 6b, c. Around 44% and 67% light transmission is achieved for Mo and Au modification layers (~10 nm) at 635 nm, respectively

(Supplementary Fig. 6b). The measured photovoltage is comparable with light illumination (0.95 W/cm$^2$) from the front and the back (Supplementary Fig. 6c), which is consistent with results shown in Fig. 2e that desirable photovoltage can be sustained even with a certain level of light absorption as long as there is sufficient light intensity. These results suggest the feasibility of using the Mo modified Si

diode as an effective optoelectronic interface for in vivo neural modification.

The degradation properties of Si-based optoelectronic devices are evaluated in phosphate-buffered saline (PBS, pH 7.4) under accelerated conditions (65 °C). The photographs of the degradation process at different stages are shown in Fig. 2j. The results indicate that the metallic films dissolve first, while Si membranes gradually disintegrate and dissolve along with the swelling and hydrolysis of the polymeric PLLA-PTMC substrate, leading to complete degradation of the entire device in about 6 weeks.

## Properties of interface modification layers

To further elucidate the characteristics of the interface modification layer on the photoresponse of the devices, we evaluate the cyclic voltammetry (CV) characteristics, impedance, and surface roughness of Mo and Au modification layers on the $p^+$ side of the Si diode ($p^+n$). The CV measurement is performed based on a 3-electrode configuration (Supplementary Fig. 7a). Mo or Au modified $p^+$ surface of the Si-diode serves as the working electrode, PBS (pH 7.4) serves as the electrolyte, and platinum (Pt) and silver/silver chloride (Ag/AgCl) are used as the counter electrode and reference electrodes respectively. The CV results suggest a significant increase in capacitance is associated with Mo modification layer (Fig. 3a and Supplementary Fig. 7). Mo modified surface also exhibits a low impedance at the lower range of frequency compared to the Au decorated or bare Si surface (Fig. 3b), indicating a lower charge transfer resistance at the device/solution interface. An equivalent circuit is proposed to analyze the impedance characteristics of the Mo and Au modified interface (Supplementary Fig. 8a), and the interface of Si/metallic film and metallic film/solution are considered as a parallel RC circuit, respectively. The fitting parameters in Supplementary Fig. 8b, c reveal that the charge transfer resistance at the metallic film/solution interface of Si/Mo devices (1480 Ω) is much lower compared with that of Si/Au devices (19604 Ω), probably due to more significant electrochemical reactions at the interface as Mo has multiple oxidation states, enabling efficient charge

injection. In addition, the capacitance of Si/Mo devices ($9.18 \times 10^{-4}$ F) is greater compared with that of Si/Au ($6.31 \times 10^{-6}$ F), consistent with the CV results, which could be attributed to the pseudocapacitive behavior associated with potential electrochemical reactions of Mo[74]. X-ray photoelectron spectroscopy (XPS) analysis is widely adopted for surface analysis of metallic thin films[75,76]. The XPS measurement of the surface of the Mo modification layer demonstrates slightly increased amounts of $Mo^{4+}$ and $Mo^{6+}$ after CV cycling (Fig. 3c and Supplementary Fig. 9a), suggesting the presence of potential oxidation. Moreover, the enhanced surface roughness of Si/Mo and Si/Au devices can also contribute to increased capacitance (Fig. 3d and Supplementary Fig. 9b, c). Based on these results, we speculate that the Mo decoration layer at the $p^+$ side can potentially allow stronger electrochemical reactions at the device/solution interface and induce pseudocapacitive behavior, which is beneficial for efficient charge injection by improving both faradaic and capacitive contributions. Considering charge conservation, corresponding charge injection at the n side is expected to increase as well, promoting charge injection and thereby stimulation efficacy, through photocapacitive effects and/or associated electrochemical reactions. For example, reactive oxygen species (ROS) could be produced at the interface that modulates neural activity[52,77]. Nevertheless, the amount of potentially generated ROS could be minimal and was not detected in the current experiments.

## Sciatic nerve modulation

As the biodegradable Mo modification layer can significantly promote photoresponse of Si diodes, we first study the modulation of the sciatic nerve of SD rats with open surgery to evaluate the stimulation efficacy of the biodegradable optoelectronic device, by monitoring the acute response upon light illumination (Fig. 4a). During our experiments, we record compound muscle action potential (CMAP) of gastrocnemius muscles using dual muscle electrodes and capture limb movement displacements using a video camera. The flexible neural interface enables conformal contact on the sciatic nerves (Fig. 4b), without sutures or bio-glues. Irradiating

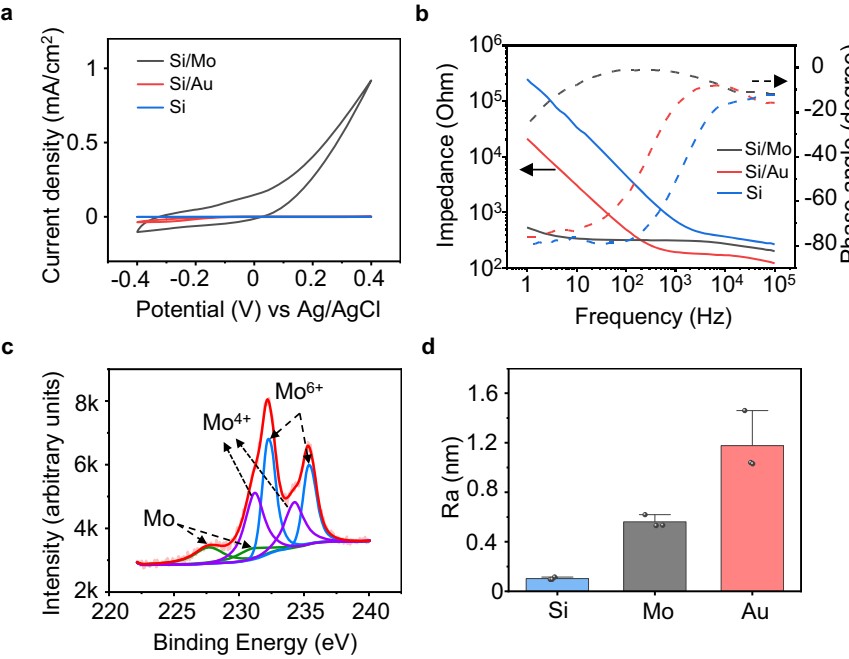

**Fig. 3 | Characteristics of interface modification layers of Si-based optoelectronic devices. a** Representative cyclic voltammetry (CV) curves of Si/Mo, Si/Au, and Si devices. **b** Representative Bode plots of electrochemical impedance (EIS) at the $p^+$ side of Si/Mo, Si/Au, and Si devices (electrode area: 1 cm$^2$). **c** Representative X-ray photoelectron spectroscopy (XPS) analysis results of the Mo modification layer surface of Si/Mo devices after 10 cycles of CV (−0.4 V−0.4 V). **d** Surface roughness of the modification layers (Mo or Au) or the $p^+$ side of the Si surface (no modification). In **a**–**d**, $n = 3$ in each group. All data are presented as mean ± s.e.m.

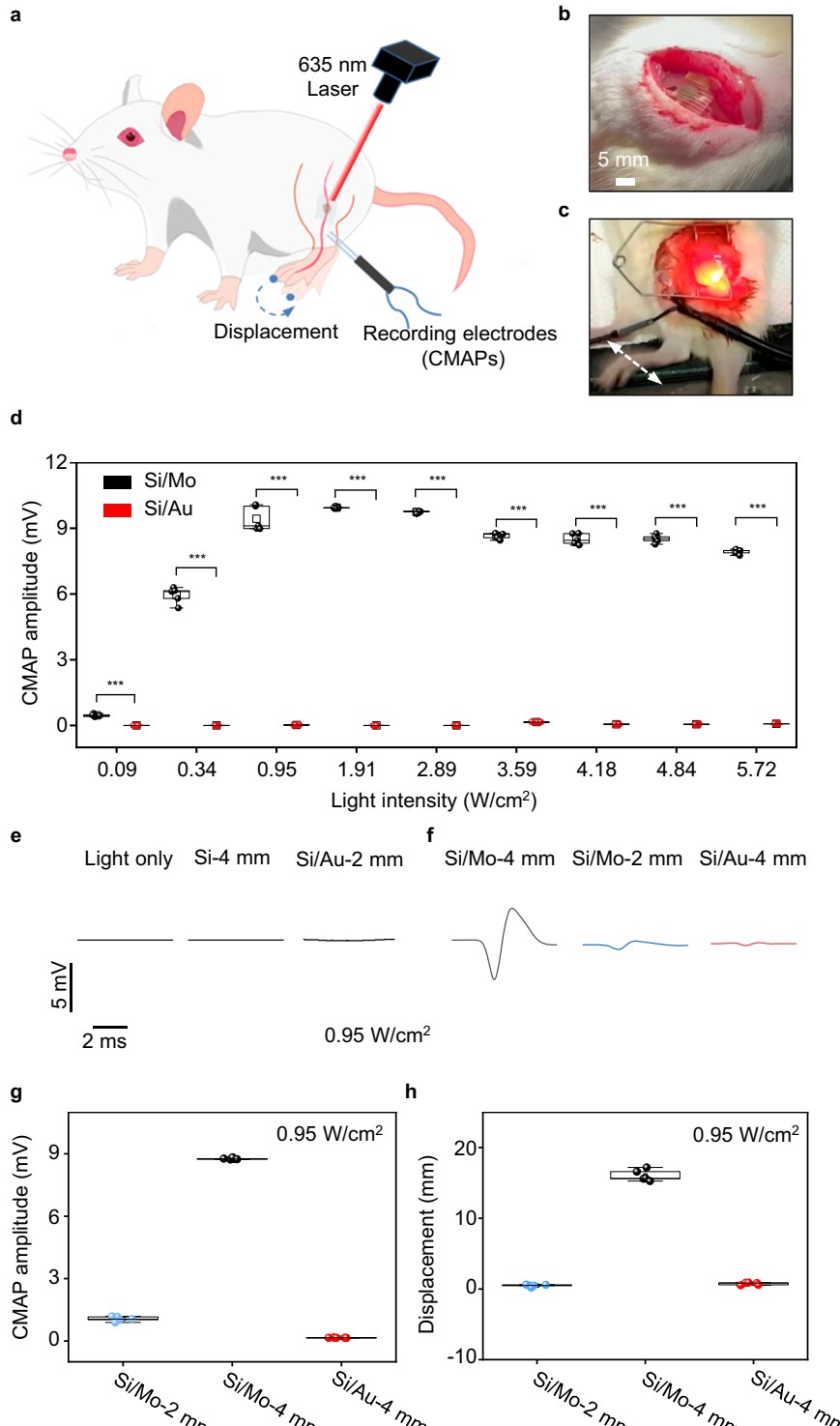

**Fig. 4 | Optoelectronic stimulation of sciatic nerves of SD rats with Si-based neural interfaces. a** Schematic illustration of a Si-based optoelectronic device attached to the sciatic nerve of the right hind limb of the rat for optoelectronic stimulation. Electromyography (EMG) recording electrodes are inserted into the gastrocnemius muscles of the right hind limb associated with the sciatic nerve to record the compound muscle action potential (CMAP). **b** Photograph of the implantation of a biodegradable silicon-based optoelectronic device on to the sciatic nerve. **c** Photograph of the movement of the right hind limb of the rat induced by pulsed laser illumination (635 nm, 0.95 W/cm², 10 Hz, pulse width 10 ms). **d** Recorded CMAP amplitude (peak-peak value) from optoelectronic stimulation of the sciatic nerve with Si/Mo and Si/Au devices (4 mm in diameter) under various light intensities. **e** Direct laser irradiation (0.95 W/cm²), Si device (no

interface modification) and Si/Au devices (2 mm in diameter) under laser illumination (0.95 W/cm²) elicit negligible response of the sciatic nerve. **f** Representative evoked CMAP induced by laser irradiation (0.95 W/cm²) on different devices: Si/Mo devices (4 mm or 2 mm in diameter) and Si/Au devices (4 mm in diameter). **g, h** CMAP amplitude (peak-peak value) and limb displacement induced by optoelectronic stimulation (0.95 W/cm²) of Si/Mo devices (4 mm or 2 mm in diameter) and Si/Au devices (4 mm in diameter). In **b**–**h**, $n = 5$ independent devices for each groups. All data are presented as mean ± s.e.m. Statistics is analyzed through SPSS (version 23.0), followed by one-way ANOVA (* $p < 0.05$, ** $p < 0.01$, *** $p < 0.001$). The box plot presents the median (center line), lower quartile (lower border), upper quartile (upper border), maximum (upper whisker) and minimum (lower whisker), which are ≤ 1.5 times the interquartile range.

the sciatic nerves with a red laser beam (635 nm, 10 Hz, 10 ms pulse width, spot size of ~2 mm) induces their excitation, leading to the contraction of associated muscles triggering movement in the right hind limb (Fig. 4c and Supplementary Movie 1). We investigated the CMAP amplitude (peak-peak value) of gastrocnemius muscles which are innervated by the sciatic nerve, under varying light intensities using the 4 mm diameter optoelectronic device (Fig. 4d). The results demonstrated that CMAP amplitudes increased with higher light intensities, reaching a plateau at 0.95 W/cm$^2$. This correlation is consistent with the patch-clamp measurement (Fig. 2e). Furthermore, the CMAPs amplitude and limb movement displacements induced by Si/Mo devices are significantly greater than those of Si/Au devices at the same light intensity (Fig. 4d, Supplementary Mov. 1), indicating a highly efficient neural stimulation triggered by the Si/Mo device. By contrast, direct laser irradiation or Si device without interface modification can barely activate the sciatic nerve (Fig. 4e). It is noted that successful stimulation of sciatic nerves with Si devices has been achieved in C57BL/6 mice[53], possibly due to the lower threshold voltage for sciatic nerve stimulation in C57BL/6 mice compared with SD rats used in the current study. With improved stimulation efficacy by Mo decoration, nerve activation can be accomplished on the sciatic nerve of SD rats. We also study neural stimulation using devices with smaller sizes (2 mm in diameter), and the evoked CMAPs and measured photovoltage with different illumination intensities are given in Fig. 4f and Supplementary Fig. 10, respectively. Although a decrease in photoresponse is observed with smaller device sizes, Si/Mo devices (2 mm) can still elicit CMAP, suggesting a potential increase in spatial resolution for neuromodulation. By contrast, Si/Au devices (2 mm) are unable to activate the sciatic nerve (Fig. 4e). The CMAP amplitudes and associated limb displacements of Si/Mo and Si/Au devices under laser irradiation (0.95 W/cm$^2$) are summarized in Fig. 4g and h respectively. The results suggest that the presence of Mo modification layer enables CMAP amplitudes up to ~9 mV and limb displacement up to ~16 mm. The achieved limb movement surpasses previously reported Si-based flexible optoelectronic devices under similar light intensity[52,53], indicating improved stimulation efficacy. This is probably due to the efficient charge injection induced by Mo decoration by improving both the faradaic and capacitive contribution. Photovoltage is produced in the Si diode under illumination via photovoltaic response, and electrons accumulated at the n side of the Si diode result in cell depolarization and initiate action potentials through photocapacitive and/or photoelectrochemical effects, while holes on the p$^+$ side of the Si diode are transported through the modification layer and extended electrodes to complete the current path. The presence of Mo modification layer could allow more effective charge injection through associated electrochemical reactions, enabling significantly improved stimulation efficacy.

## Transdermal facial nerve modulation and regeneration

As Mo modified devices demonstrate desirable stimulation efficacy, which could potentially enable transdermal neural modulation with fully biodegradable devices in larger animals, we further study the feasibility of transdermal biomodulation of facial nerves of New Zealand rabbits. Compared to sciatic nerves, facial nerves are located closer to the skin surface, resulting in reduced absorption and scattering of penetrating light, greatly facilitating the successful implementation of transdermal optoelectronic stimulation. Facial nerve injuries often occur during surgery in the head region[78], and can lead to paralysis of the facial muscles and cause facial palsy. Optoelectronic stimulation could potentially provide electrical cues to facilitate tissue regeneration and functional restoration[79,80]. Although facial nerves are more accessible to tissue-penetrating light compared to sciatic nerves, there has rarely been attention to the use of optoelectronic stimulation in facial nerve phototherapy, probably due to the unavailability of

biodegradable miniaturized optoelectronic devices with optimal stimulation efficacy[81,82]. We first investigate the evoked CMAP amplitudes of the facial nerves of New Zealand rabbits in comparison to the sciatic nerves of SD rats under different stimulation currents using conventional electrodes (Supplementary Fig. 11a). The results suggest that, compared to the sciatic nerve of SD rats, the facial nerve of New Zealand rabbits produces weaker muscle responses at the same stimulation current intensity. This is likely because the sciatic nerve in SD rats innervates a much larger muscle group with a greater volume[83,84]. As absorption and scattering can reduce light intensity when penetrating through tissues[85], we study the photoresponse under transdermal illumination conditions by placing a skin sample between incident light and the optoelectronic device (Fig. 5a, b). We monitor the output light intensity penetrated through the skin sample (Fig. 5c), and the results demonstrate a ~50% reduction in light intensity. The measured photovoltage of the Si/Mo devices through the skin is reduced by ~30% due to the decrease in penetrated light intensity caused by the absorption of the skin, separated extended electrodes, and the modification layer (Fig. 5d).

As shown in Figs. 5e and 5f, the flexible optoelectronic device enables conformal contact with the facial nerve, ensuring a desirable device/nerve interface. When subjected to laser irradiation (635 nm, 10 Hz, 10 ms pulse width, 2 mm spot size, 0.95 W/cm$^2$), the Si and Si/Au devices barely elicit activation of the facial nerve even without skin coverage (Fig. 5g). By contrast, the Si/Mo devices successfully induce CMAPs under transdermal illumination (Fig. 5g). It is noted that the measured CMAP amplitude is much smaller compared to that measured on the sciatic nerve of SD rats, consistent with the results presented in Supplementary Fig. 11a. We investigate the biodegradability of the Si/Mo optoelectronic devices after implanted at the facial nerve (Fig. 5h). The device maintains its integrity and remains in contact with the nerve after implantation of 4 days (Fig. 5h and Supplementary Fig. 11b), and the photovoltage of the retrieved device remain ~30% of the initial photoresponse after 6 days of implantation (Supplementary Fig. 11c). The PLLA-PTMC polymeric substrate of the devices exhibits great reduction in size through hydrolysis after 4 weeks of implantation (Fig. 5h), and the device completely degrades after 8 weeks of implantation, indicating full biodegradation in vivo (Fig. 5h).

With desirable transdermal photoresponse and biodegradability of Si/Mo devices, we further investigate potential optoelectronic therapy for tissue repair and function restoration of facial nerves, as research has demonstrated that electrical cues at the initial stage (first few days) after injury can considerably accelerate the process of peripheral nerve regeneration[54,61,86]. As approximately 60% of facial nerve injuries are non-truncated (contusions, compressions, partial avulsions)[87], we investigate facial nerves with crush injuries and assess the efficacy of neural stimulation using the biodegradable optoelectronic device to promote functional recovery.

The rabbit facial tissue is carefully dissected to fully expose the facial nerve, followed by the creation of a crush injury (Supplementary Fig. 11d). The fully biodegradable and flexible Si/Mo optoelectronic device is conformally attached to the injured site, followed by suturing of the skin tissue and demarcating the implantation site by makers of surgical sutures. Phototherapy is applied 1-h daily for 1 or 4 consecutive days, with transdermal irradiation (635 nm, 10 Hz, 10 ms pulse width, 0.95 W/cm$^2$, spot size ~2 mm). The average power density of the pulsed laser (0.19 W/cm$^2$) employed in the current studies falls within the skin exposure limit (0.2 W/cm$^2$)[88], suggesting minimal adverse effects. To evaluate the efficacy of optoelectronic stimulation, studies are performed for 5 groups: (i) implantation of Si/Mo devices with optoelectronic stimulation for 4 days (Si/Mo OS-4d); (ii) implantation of Si/Mo devices with optoelectronic stimulation for 1 day (Si/Mo OS-1d); (iii) implantation of Si devices (no modification layer) with optoelectronic stimulation for 4 days (Si OS-4d); (iv) Photostimulation without optoelectronic devices for 4 days (PS-4d); and (v) no treatment (control).

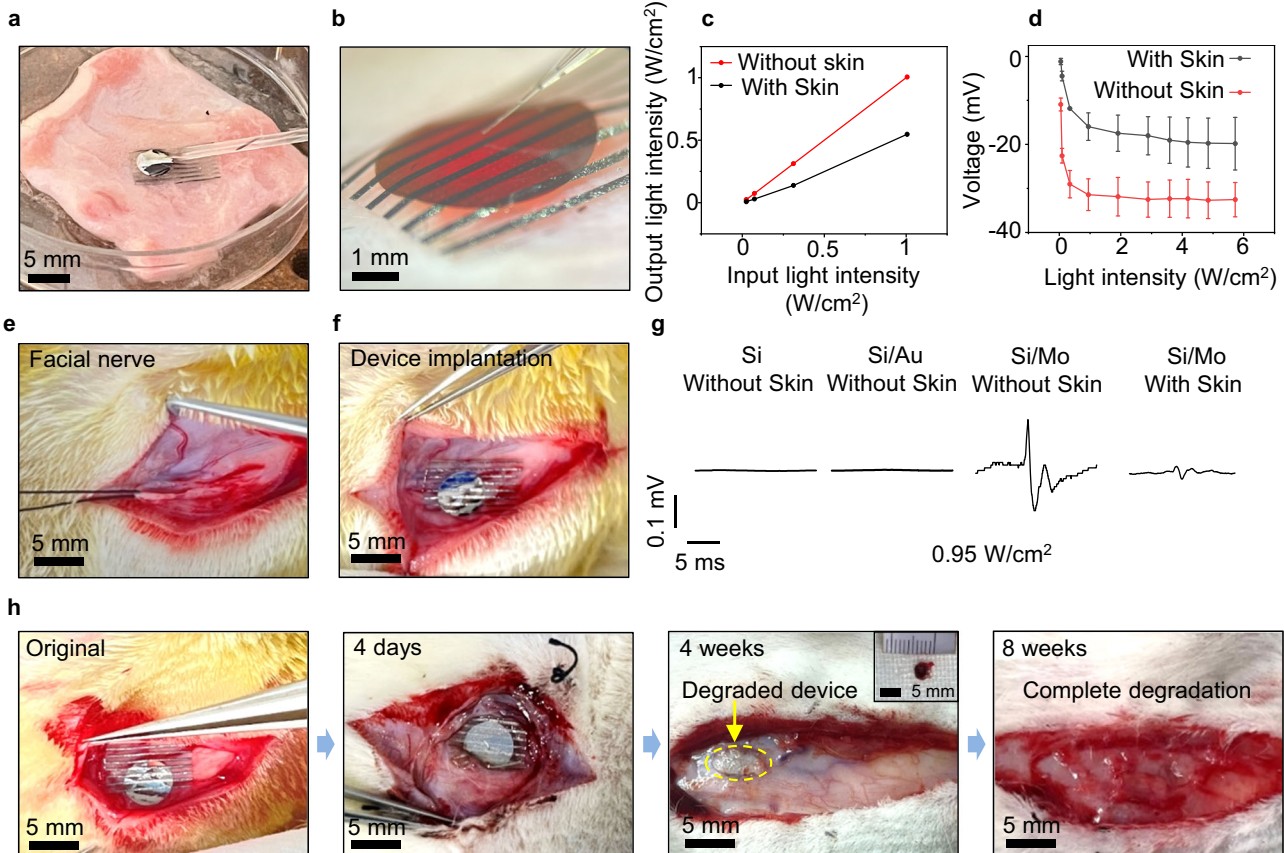

**Fig. 5 | Optoelectronic stimulation of facial nerve activity in New Zealand rabbits with Si-based neural interfaces. a, b** Images of the patch-clamp set-up to investigate the photoresponse of Si/Mo devices under transdermal irradiation (facial skin, ~ 2 mm). Laser is incident from the back and passes through the skin. **c** The measured output light intensities under the illumination with different intensities (635 nm, 10 Hz, pulse width 10 ms) with/without skin coverage. **d** Measured photovoltage of Si/Mo devices with/without the presence of skin coverage under illumination with different intensities (635 nm, 10 Hz, pulse width 10 ms). **e** Anatomy of the facial nerve in the New Zealand rabbit. **f** Photograph of the implantation of a device on the facial nerve. **g** Evoked CMAPs of Si, Si/Au and Si/Mo devices under laser irradiation (635 nm, 10 Hz, pulse width 10 ms, 0.95 W/cm²) with/without facial skin coverage. **h** Images of the in vivo degradation of the Si/Mo optoelectronic device at various stages. The inset at 4 weeks shows a retrieved partially degraded device. In **a**, **b**, **d**, $n = 10$ independent experiments. In **e–h**, $n = 4$ independent experiments. All data are presented as mean ± s.e.m.

Regenerated nerve segments are harvested at 4 weeks and 8 weeks postoperative, and are subjected to tissue sectioning and immunofluorescence staining to evaluate tissue regrowth. The immunofluorescence staining images of the longitudinally sectioned nerve segments located 5 mm distal from the crush sites at 4 weeks postoperatively suggest that, greater number of directional axons denoted by both GAP-43 and NF200 (green fluorescence) is observed in the Si/Mo OS-4d group compared with the other groups (Fig. 6a, b and Supplementary Fig. 12). Moreover, the density of myelin sheaths denoted by MBP (red fluorescence) surrounding the nerve fibers (NF200, green fluorescence) is also greater in the Si/Mo OS-4d group compared to the other groups (Fig. 6a and Supplementary Fig. 12a). Furthermore, the immunofluorescence staining images of transversely sectioned nerve segments located 10 mm distal from the crush sites at 8 weeks postoperatively (Fig. 6c) indicate that, myelination in the Si/Mo OS-4d and Si/Mo OS-1d groups is more mature than the other groups, in the sense that the presence of fluorescent signals of myelin sheaths (MBP, red fluorescence) are greater surrounding the nerve fascicles regions, indicating a more profound and expedited process of myelination. These results provide evidence of the beneficial effects of the biodegradable optoelectronic neural interface for facial nerve regeneration. In addition, hematoxylin and eosin (H&E) staining images are illustrated in Supplementary Fig. 13 (4 weeks) and Supplementary Fig. 14 (8 weeks), indicating no significant inflammation.

To evaluate the restoration of nerve conduction function, a two-electrode current stimulator is positioned proximal to the site of nerve injury at 4 weeks and 8 postoperatively. A single current stimulation of 3 mA (pulse width 2 μs) is subsequently applied using electrophysiological equipment, and the resulting CMAPs generated at the target muscle are recorded. The amplitude and latency of CMAPs are assessed as a measure of nerve conduction recovery. The CMAP data at 4 weeks postoperatively demonstrate relatively large variation, probably due to incomplete tissue regeneration and re-innervation to the target muscles at such an early stage. Nevertheless, more instances of greater CMAP amplitudes are observed in the Si/Mo OS-4d and Si/Mo OS-1d groups, suggesting a greater chance of faster nerve re-innervation. Statistically, the measured CMAP amplitude (peak-peak value) at 4 weeks postoperatively is significantly greater in the Si/Mo OS-4d and Si/Mo OS-1d groups ($p < 0.01$ or $p < 0.05$) as compared to the other groups (Supplementary Fig. 15a). The latency of the Si/Mo OS-4d and Si/Mo OS-1d groups are lower compared to the Si OS-4d group ($p < 0.001$), and the latency of the Si/Mo OS-1d group is lower than that of the Control and PS-4d groups ($p < 0.05$) (Supplementary Fig. 15b). At 8 weeks postimplantation when nerve regeneration has progressed to a more mature stage, the Si/Mo OS-4d group exhibits significantly greater amplitude compared to the control, PS-4d, Si OS-4d and Si/Mo OS-1d groups ($p < 0.001$) (Fig. 6e), and the amplitude is approaching that of the normal side (wild-type, WT). The Si/Mo OS-1d group also demonstrates significantly greater CMAP amplitude

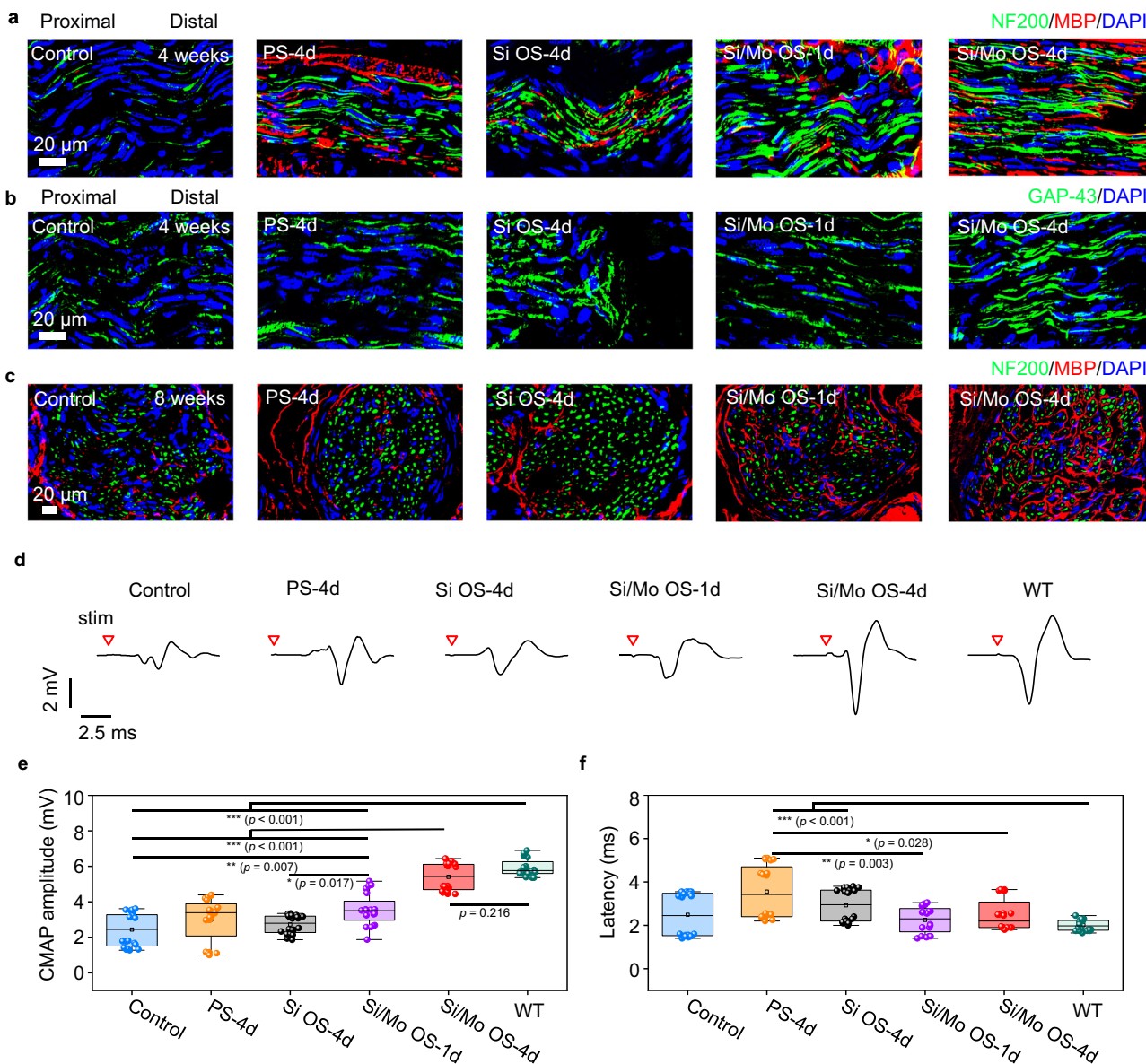

**Fig. 6 | Evaluations of regenerated nerve segments at 4 and 8 weeks after phototherapy with the biodegradable optoelectronic neural interfaces.**
**a**, **b** Immunofluorescence staining images of longitudinally sectioned regenerated nerve segments located 5 mm distal from the crush sites at 4 weeks post-operatively. Phototherapy is applied 1-h daily for 1 or 4 days (635 nm, 10 Hz, pulse width 10 ms, 0.95 W/cm²). Groups: Control (no treatment), PS−4d (photostimula-tion for 4 days), Si OS-4d (optoelectronic stimulation for 4 days with Si devices), Si/Mo OS-1d (optoelectronic stimulation for 1 day with Si/Mo devices), Si/Mo OS-4d (optoelectronic stimulation for 4 days with Si/Mo devices). **a** Immunohistochemical staining: axons (NF200, green), myelin sheaths (MBP, red), and nuclei (DAPI, blue). **b** Immunohistochemical staining: newly grown axons (GAP-43, green), and nuclei (DAPI, blue). **c** Immunofluorescence staining images of transversely sectioned

regenerated nerve segments located 10 mm distal from the crush sites at 8 weeks postoperatively. Immunohistochemical staining: axons (NF200, green), myelin sheaths (MBP, red), and nuclei (DAPI, blue). **d** Representative evoked CMAPs at 8 weeks postoperatively. Red triangle indicates the point of light stimulation. **e** CMAP amplitude at 8 weeks postoperatively. **f** Latency of CMAPs at 8 weeks postoperatively. In **a**–**f**, $n = 4$ independent experiments. All data are presented as mean ± s.e.m. Statistics is analyzed through SPSS (version 23.0), followed by one-way ANOVA (* $p < 0.05$, ** $p < 0.01$, *** $p < 0.001$). The box plot presents the median (center line), lower quartile (lower border), upper quartile (upper border), max-imum (upper whisker) and minimum (lower whisker), which are ≤ 1.5 times the interquartile range.

compared with the control ($p < 0.01$) and Si OS-4d groups ($p < 0.05$). No significant difference in latency is observed among the Si/Mo OS-4d and Si/Mo OS-1d groups compared to the WT, control and Si OS-4d groups (Fig. 6f). The CMAP and immunohistochemical results suggest that the Mo modified optoelectronic device can efficiently promote reinnervation following nerve injuries.

The enhancement of long-term axonal growth through elec-trical stimulation in the early recovery phase may be attributed to the expedited Wallerian degeneration[62] and the prompt and continuous upregulation of regeneration associated genes[60,89],

which accelerates staggered regeneration of axons across the injury site and therefore facilitates more rapid axonal regrowth. We assess the degree of degeneration of myelinated axons 7 days postimplantation of the control and Si/Mo OS-4d groups. Tolui-dine blue staining and TEM analysis of the transversely sectioned nerve segments, located 10 mm distal from the crush site on day 7 postoperatively are given in Supplementary Fig. 16. In the control group, a substantial amount of degenerating myelinated axons (marked by the yellow arrow) and some intact myelinated axons (marked by the red arrow) are observed. In contrast, in the Si/Mo

OS-4d group, axonal degeneration is more advanced, evidenced by the greater clearance of myelin and axons, resulting in significantly fewer degenerating myelinated nerve fibers and few intact axons. The statistical results show significantly less density of myelinated nerve fibers (including both degenerating and intact axons) in the Si/Mo OS-4d group compared with the control group ($p < 0.01$), based on the toluidine blue staining images. The results are consistent with the previous report[62] suggesting a 4-day optoelectronic stimulation at the early phase could accelerate Wallerian degeneration, and thereby creating a more favorable microenvironment for nerve regrowth. Collectively, these results suggest that the fully biodegradable Si based optoelectronic device with Mo decoration can effectively modulate nerve activity and facilitate functional recovery by providing electrical cues at the initial state of nerve injuries.

## Discussion

Herein, we propose a fully biodegradable, flexible and miniaturized Si-based neural interface that achieves transdermal optoelectronic stimulation and regeneration of peripheral nerves. Dissolvable Mo decoration at the device interface enables significantly enhanced charge injection and efficient neural stimulation. Successful activation of the sciatic nerve in SD rats and the facial nerve in New Zealand rabbits is demonstrated. Furthermore, accelerated tissue repair and functional recovery of injured facial nerve is accomplished in New Zealand rabbits through transdermal optoelectronic stimulation. The entire device is bioresorbable in vivo, eliminating the need for retrieval surgery and, therefore, minimizing associated infection risks. Further studies of encapsulation strategies could enable a more controllable operational time frame. Combined with a more sophisticated device structure with extended leads or implantable fibers for light delivery, optoelectronic stimulation at deeper tissue may be possible. Applicable scenarios could also be extended to other nervous systems with proper device optimization, including the vagus nerve, spinal cord, retinal neurons, etc. Overall, this work offers materials strategies and device schemes to achieve highly efficient and biodegradable optoelectronic neural interfaces for non-genetic, wireless, and transdermal neural modulation and regeneration, with potential use in clinical medicine for maximized therapeutic outcomes.

## Methods

### Device fabrication

PLLA-PTMC films are prepared by drop casting. The PLLA-PTMC solution is obtained by dissolving PLLA-PTMC particles (60:40 Jinan Daigang Biomaterials Co., Ltd., China) in chloroform (analytical pure, Beijing Tongguang Chemical Co., Ltd., China) at a mass/volume ratio of 1:10. Then 4 mL of the solution is drop-casted onto glass substrates and cured at 4 °C for 12 h. After solvent evaporation, PLLA-PTMC films (thickness ~ 200 μm) are obtained. Shadow masks with the pattern of electrode arrays (line width 300 μm, pitch 700 μm) are then adhered to the PLLA-PTMC surface, followed by deposition of extended electrodes (Mo or Au, 300 nm) using magnetron sputtering.

SOI wafers (device layer: n-type, 100 crystal orientation, resistivity 1–10 Ω·cm, thickness 2.5 μm) were purchased (Beijing Topvendor, China). Vertical Si diodes (p⁺n) are achieved by ion implantation. Boron (B) was implanted into the n-type Si (doping concentration of $4 \times 10^{14}$/cm²) under injection energy of 30 keV and then annealed at 950 °C for 30 min for dopant activation. Si diodes are patterned into miniaturized sizes (2 mm or 4 mm in diameter) by photolithography and reactive ion etching. The buried oxide layer in the SOI is removed using HF solution (concentration 40%, ACS grade, Aladdin, China) and thin film Si diodes (2.5 μm) are released. Next, modification layers (Mo or Au, 10 nm) are deposited on the p⁺ side of Si diodes using magnetron sputtering, and then modified Si diodes are transferred to the surface of the PLLA-PTMC substrates with deposited extended electrodes

using a hot release tape, where the modified p⁺ side of the Si diodes is in contact with extended electrodes which functions as return electrodes.

### Optoelectronic characterization of the device

The device is immersed in a PBS solution, and a standard patch-clamp system is used to measure the photoresponse of the optoelectronic device. A red laser beam (635 nm laser, 10 Hz, pulse width of 10 ms, spot size of 2 mm) is incident on the device through a collimating lens, and its irradiation intensity is controlled by managing the input current of a laser diode. Voltage- and current-clamp protocols are performed by a p-Clamp software-controlled (Molecular Devices) Axonpatch 200B amplifier. A glass pipette (~ 1 MΩ) loaded with PBS solution approached the device surface within a distance of 5 μm. And voltage- or current- clamp (filtered at 10 kHz and sampled at 200 kHz) is used to record the transient photoresponse of the device. We measure the photovoltage of Si diodes without modification layers (Si devices), Si diodes with a biodegradable Mo modification layer (Si/Mo devices), and Si diodes with a traditional Au modification layer (Si/Au devices). It is noted that extended electrodes are not present for devices without modification layers (Si devices).

### Characterization of device degradation

Degradation experiments were conducted by immersing the Si/Mo device in PBS solution (pH 7.4), with daily replacement of the PBS solution, which is maintained at 65 °C using a water bath. Optical images were recorded at various stages.

### Electrochemical characterizations of the optoelectronic device

CV and EIS tests were performed using a potentiostat (CHI650, Chenhua, China). A three-electrode system was adopted. PBS served as the electrolyte, the Si diode (p⁺n, on SOI) with or without modification layers served as the working electrode (with an area of 1 cm²), Ag/AgCl was used as the reference electrode, and the mesh Pt electrode (with an area of 1 cm²) was used as the counter electrode. 300 nm of Au is sputtered on the top part of the Si diode as the electrical contact for measurement. CV curves were measured at a scan rate of 0.025 V/s and a potential range from -0.4 V to 0.4 V. EIS measurements were performed from 1 Hz to 0.1 MHz.

### Characterization of material morphology

The surface morphology of the material was observed by field emission scanning electron microscopy (SEM) (Zeiss, Berlin, Germany) and the elemental composition was analyzed by energy spectrometry (EDS). X-ray photoelectron spectroscopy (XPS) (250XI, Thermo Fisher, England) was used to analyze the surface chemistry of Mo modification layer. A Bruker Dimension Icon microscope with a ScanAsyst probe in automated peak force tapping mode was used for atomic force microscopy (AFM) characterization. Nano scope analysis 1.9 software was used to remove scars from the scans, crop the scan area and plot the three-dimensional morphology of the material surface.

### Characterization of photothermal effects

An infrared thermal imaging camera (220 S, Fotric, China) was used to monitor the temperature change of the device after illumination. The facial skin of rabbits was placed in a transparent glass dish with the appropriate amount of saline, and Si/Mo device was placed on the skin. Pulsed light (635 nm laser, 10 Hz, pulse width of 10 ms, spot size of 2 mm) is incident from the back side of the skin, and the temperature change is recorded by the camera.

### Acute sciatic nerve stimulation

All procedures associated with the animal studies are in accordance with the institutional guidelines of the Chinese PLA General Hospital,

Beijing, China. The experimental protocol was reviewed and approved by the institutional animal care and use committee at the Chinese PLA General Hospital, Beijing, China (approval number 2016-x9-07). All SD rats were anesthetized through the intraperitoneal injection of a 1% solution of sodium pentobarbital at a dose of 0.3 ml/100 g body weight, and the hair on the right femur was subsequently removed. An incision was made across the midline of the skin, and the fascial plane was opened between the gluteus maximus and the anterior head of the biceps femoris, thereby exposing the sciatic nerve. Attaching the optoelectronic devices to sciatic nerves, and parallel recording electrodes were placed on the gastrocnemius muscle to record CMAPs. The movement of the right hind limb is recorded by a camera.

### Acute facial nerve stimulation

The New Zealand rabbits were weighed and subsequently anesthetized through the intramuscular injection of a combination of Zoletil 50 and Xylazine (Zoletil + Xylazine) at a dose of 0.8 ml/kg. An incision was made in the right cheek and the subcutaneous fascia was separated to isolate the nerve from the surrounding tissue and blood vessels. The buccal branch of the facial nerve was then exposed and the optoelectronic device was attached to the nerve. Then parallel recording electrodes were placed on the target muscle in the face to record the CMAPs evoked by the optoelectronic device.

### Functional restoration of facial nerve injury in rabbits

The experimental protocol received approval from the Experimental Animal Ethics Committee of PLA General Hospital (approval number: 2016-x9-07). All animal-related procedures were conducted in line with the institutional guidelines of PLA General Hospital. Forty New Zealand Large White rabbits (no distinction between male and female), aged 3 months and weight between 2 and 2.2 kg, were selected for the study. The rabbits were subsequently anesthetized through the intramuscular injection of a combination of Zoletil 50 and Xylazine (Zoletil + Xylazine) at a dose of 0.8 ml/kg. The right face of the animal was disinfected with iodophor in preparation for all surgical procedures, which were performed under aseptic operating conditions by two surgeons. An incision was made in the right cheek and the subcutaneous fascia was separated to isolate the nerve from the surrounding tissue and blood vessels. The procedure involved exposing the buccal branch of the facial nerve and inducing crush injuries by clamping the nerve twice with toothless hemostatic forceps for 30 seconds, with 10-second intervals between clamps. The hemostat's ring handles have a ratcheted locking mechanism that secures the jaws in a fixed position. We utilize the same position of the jaws of the hemostat to create crush injuries and employ the ratchet mechanism to lock the jaws in identical positions, ensuring consistent force and therefore reproducible degree of injuries for each experiment. Then the optoelectronic devices were implanted and attached to the injured nerve. After the procedure, the wound was closed sequentially, disinfected and the rabbits were fed normally. Phototherapy was applied 1-h daily for 1 or 4 days (635 nm, 10 Hz, pulse width 10 ms). The following groups are investigated ($n = 4$): Control (no treatment), PS-4d (photostimulation for 4 days), Si OS-4d (optoelectronic stimulation for 4 days with Si devices), Si/Mo OS-1d (optoelectronic stimulation for 1 day with Si/Mo devices), Si/Mo OS-4d (optoelectronic stimulation for 4 days with Si/Mo devices).

### Electrophysiology assessment

At 4 and 8 weeks postimplantation, rabbits were weighed and anesthetized through the intramuscular injection of a combination of Zoletil 50 and Xylazine (Zoletil + Xylazine) at a dose of 0.8 ml/kg. A complete dissection was conducted to expose both the right facial nerve bridge and the normal facial nerve on the left side. Synergy electromyography (Oxford, USA) stimulating electrodes were placed on the proximal nerve stump, and parallel recording electrodes were placed on the facial target muscle. The stimulation current was set to 3 mA (pulse width ~ 2 μs), and CMAPs were recorded for a single stimulation. CMAPs were also measured on the left side with normal functions which are denoted as the WT group. The delay time (latency time) and peak–peak value (amplitude) of the CMAP on the injured/unoperated side were analyzed and derived for each group ($n = 4$). Latency time: the period from the electrical stimulation to the onset of the first significant H&E signal peak. Amplitude: the value of the peak minus the trough in a complete waveform.

### Immunohistochemical and historical assessment of nerve tissues

At weeks 4 and 8 postoperatively, animals were sacrificed via intramuscular injection of an overdose of Zoletil 50 and Xylazine (Zoletil + Xylazine) in each group. The regenerated nerve segments were immediately removed, fixed in 4% paraformaldehyde (ten times the sample volume) at 4 °C for 24 h, dehydrated in sunken sugar, embedded in optimal cutting temperature compound gel, and snap-frozen in liquid nitrogen. The nerve grafts were cut into longitudinal sections (week 4) or transverse sections (week 8) with a thickness of 10 μm. The samples were randomly divided into two groups: one for immunofluorescent staining (using NF-200, S-100, GAP-43, MBP and DAPI) and the other for H&E staining ($n = 4$). Transversely sectioned nerve segments on day 7 postoperatively were evaluated with toluidine blue (TB) staining and transmission electron microscopy (TEM) ($n = 3$).

### Statistical data

Data are presented as mean ± standard deviation. Data were statistically analyzed by the SPSS software package (version 23.0), accompanied by one-way analysis of variance (ANOVA) (* $p < 0.05$, ** $p < 0.01$, *** $p < 0.001$).

### Reporting summary

Further information on research design is available in the Nature Portfolio Reporting Summary linked to this article.

## Data availability

All data needed to evaluate the conclusions in the paper are present in the paper and/or the Supplementary Materials. Additional data related to this paper may be requested from the authors. Source data are provided with this paper.

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

## Acknowledgements

The project was supported by the National Natural Science Foundation of China (T2122010, 52171239 to L.Y., 52272277 to X.S., 32171356 to Y.W.), and Beijing Municipal Natural Science Foundation (Z220015 to L.Y.).

## Author contributions

P.S., C.Y., and L.Y. conceived and designed the research project. P.S., C.Y., J.C., Y.H., H.W., X.S., and L.Y. designed and fabricated the devices, and P.S., C.Y., L.W., X.S., and L.Y. performed data analysis. C.Y., P.S., C.L., H.H., Y.G., X.L., M.S., S.C., H.C., W.X., S.W., J.P., and Y.W. performed the animal studies. P.S., Y.M., and M.Z. designed and performed the modeling of the equivalent circuit at the interface. S.L. and K.C. contributed to the photographs and schematic diagrams in the manuscript. C.Y., P.S., and L.Y. wrote the manuscript with inputs from all authors. P.S., C.L., and C.Y. contributed equally to the paper.

## Competing interests

The authors declare no competing interests.

## Additional information

[1]School of Materials Science and Engineering, The Key Laboratory of Advanced Materials of Ministry of Education, State Key Laboratory of New Ceramics and Fine Processing, Laboratory of Flexible Electronics Technology, Tsinghua University, Beijing 100084, P. R. China. [2]Institute of Orthopedics, Chinese PLA General Hospital, Beijing Key Lab of Regenerative Medicine in Orthopedics, Key Laboratory of Musculoskeletal Trauma and Injuries PLA, No. 28 Fuxing Road, Beijing 100853, P. R. China. [3]School of Life Sciences, Tsinghua University, Beijing 100084, P. R. China. [4]Department of Electronic Engineering, Tsinghua University, Beijing 100084, P. R. China. [5]Thayer School of Engineering, Dartmouth College, Hanover, NH 03755, USA. [6]Department of Rehabilitation, Tongji Hospital, Tongji Medical College, Huazhong University of Science and Technology, Wuhan  430030, P. R. China. [7]School of Integrated Circuits, Shenzhen Campus of Sun Yat-sen University, Shenzhen 518107, P. R. China. [8]School of Biological Science and Medical Engineering, Key Laboratory of Biomechanics and Mechanobiology of Ministry of Education, Beijing Advanced Innovation Center for Biomedical Engineering, Beihang University, Beijing 100083, P. R. China. [9]School of Engineering Medicine, Beihang University, Beijing 100083, P. R. China. [10]Chinese Institute for Brain Research, Beijing 102206, P. R. China. [11]Department of Electronic Engineering, Beijing National Research Center for Information Science and Technology, Laboratory of Flexible Electronics Technology, Tsinghua University, Beijing 100084, P. R. China. [12]Institute for Precision Medicine, Tsinghua University, Beijing 100084, P. R. China. [13]IDG/McGovern Institute for Brain Research, Tsinghua University, Beijing 100084, P. R. China. [14]Co-innovation Center of Neuroregeneration, Nantong University, Nantong 226007, P. R. China. [15]MegaRobo Technologies Co. ltd, Beijing 100085, P. R. China. [16]These authors contributed equally: Pengcheng Sun, Chaochao Li, Can Yang.
✉e-mail: danica.wang@outlook.com; wangwangdian628@126.com; lanyin@tsinghua.edu.cn

