## [Peer Review File · Nature Communications]

A biodegradable and flexible neural interface for transdermal optoelectronic modulation and regeneration of peripheral nervesREVIEWER COMMENTS

Reviewer #1 (Remarks to the Author):

The authors present their work on a novel optoelectronic device for nerve stimulation. They use multiple models (species, in/ex vivo) to characterize and validate their device and demonstrate its potential for attenuating nerve injury. Materials experts may be better able to critically assess this paper since much of the paper is about materials fabrication and characterization (Figs. 2,3,5; Supplemental figs. 1 to 12, 15), not clinical medicine. However, Figure 6 does test its clinical relevance and this seems like the primary set of experiments for this paper. The preceding work does not test the use of this device on a disease/condition, but rather much of the content leading up to this was methods/device validation. Please see my detailed comments below:

The overall grammar, sentence structure, and flow could be improved.

Abstract Ln38-39: how does this differ from optogenetics? it's unclear to me how stimulation is occurring.. this does not appear to be electrical or genetic.. this information is presented later in the manuscript but would be helpful to have earlier on

Abstract Ln 44: stimulation for what purpose? pain? motor function? other? it sounds like this is 'for' regeneration of peripheral nerves... not 'and' regeneration of peripheral nerve

Pg4Ln61: new generation devices for spinal cord and peripheral nerve stimulation are much more miniaturized now

Pg4Ln62: which drug-based methods suffer from imprecise dosing? intrathecal pumps for drug delivery are quite precise

Pg5Ln75-77: include some detail about this mechanism in the opening abstract (when you contrast it to optogenetics)

Pg5Ln81: please detail examples of when device removal would be necessary and make a biodegradable system advantageous... there is often no need to remove currently available implantable neuromodulation devices

Pg5Ln89: please provide additional supporting references that electrical stimulation promotes regeneration... I see you do this later in the manuscript

Pg6Ln93: if the device is transdermal, it seems like it could be easily removed (vs implantable) and thus, make the advantage of biodegradability less clear

Pg16 Sciatic nerve modulation: the manuscript begins with a focus on transdermal application and its advantages, but this model and others presented involve direct nerve placement (with open surgery)

Pg17Ln298-300: Sentence is unclear with typos

Pg19 Ln 332-334: consider addressing some of the possible barriers or reasons why there has rarely been attention for facial nerve application

Pg19Ln337-339: this is likely due to transdermal vs direct application.. furthermore, the sciatic nerve innervates much larger muscle groups

Pg21Ln367-369: consider moving this to the intro/rationale

Pg22Ln379-386: this seems like the primary set of experiments for this paper.. the preceding work does not test the use of this device on a disease/condition.. rather much of the content leading up to this has been methods/device validation

Supplementals: can the supplemental figures be condensed or combined somehow? 20 supplemental figures are a bit much since many of these figures have small amounts of data presented

Facial nerve crush injury/S16: how is the nerve crush injury standardized for replicates? it looks like this is performed manually and would likely involve variable pressures of crush injury

Reviewer #2 (Remarks to the Author):

The present manuscript presents a flexible neural interface activated by optoelectronic transdermal stimulation. The proposed use is for electroneurostimulation following peripheral axotomy, as a method to enhance recovery of motor function after crush of the facial nerve in rabbits.

Overall, the work is well presented and organized, so that the construction of the implantable device is innovative and effective in eliciting compound muscle action potentials in a reproducible fashion. Nevertheless, the authors did not take into account the Wallerian degeneration process that takes place distally to the site of axotomy, in the present case, the site of the nerve crush. Since the severed axons degenerate post-injury, what is the role of electrostimulation? In turn, a four-day therapy is highly controversial in terms of yielding long-term axonal growth acceleration.

The immunohistochemical analysis of the nerves, shown in Figure 6, Figure S17, and Figure S19 are of below-average quality. There is a need to indicate the site of injury and the proximal and distal stumps. The increase of S100 immunolabeling appears to be linked to fibrosis and not with an enhancement of function/recovery. There is a need for a time-course evaluation of the morphological recovery, including markers of axonal regeneration, such as CGRP or GAP-43. Staining for MBP could help evaluate myelination.

The CMAP data connected with the immunolabeled material is extremely variable, showing strange groups of data within each experimental group (p. ex., Figure 6C). Infinite latency in control nerves is not acceptable.

Reviewer #3 (Remarks to the Author):

Dear P. Sun and co-authors, I have reviewed your paper concerning the development of a flexible transdermal optoelectronic device aiming to the modulation and regeneration of peripheral nerve. The overall study sounds appealing, original and tends to strengthen the hardcore research made nowadays to find a straightforward viable method to treat these diseases. The wireless method sounds intelligent and elegant indeed if compared to other sources of relatively minimally invasive techniques (RF, temporal interference, ultrasound, etc.) due to spatial resolution and other principle issues.

The handful of work shows also the importance in assessing heat transfer phenomena

during the evaluation of the mechanism of the device to decouple thermal phenomena from capacitive ones.

I kindly suggest this paper to be published in Nature Communications eventhough there are different points, in my opinion which must be definitely clarified and deeper discussed before the publication.

I would like you to compare the light intensity used from you with actual quantitative numbers of light safety limits after which the illumination is considered hazardous.

The Si/Mo modified device, seems to be so far the best solution to your scientific aim but have you performed deeper experiment to assess which are the electrochemical reactions which lead to a better performance of the device? From cyclic voltammetry is clear that Mo oxidizes in mimicking physiological solutions. What about possible oxygen reduction which will lead to reactive oxygen species production? Have you evaluated this possibility?

#In Figure S6, P. Sun and coauthors show a comparison between Si/Mo and Si/Au (10nm of decoration) regarding high retention index and I would say that the only reason for which Au fails more is due to highly likely recrystallization of the layer in Cl⁻ containing solutions which leads to faster degradation of the material .

In Figure S8 would be better to compare all the data in terms of current density and not just current because readers will be interested in the area of your samples due to couple of order of magnitude difference in current. And also please specify the Potential vs what (i.e. E(V) vs Ag/AgCl since this is the second species electrode you have used as reference)

In Figure S9 would be better to report if possible the Bode plot and not only the Nyquist in order to give an image of the phase as well. And please add the phase in Figure 3 of the actual manuscript.

In Figure S10 I would question about the XPS of qualitative data on a 10nm thick layer because what the authors are looking at could even be sputtered surface during the XPS analysis itself.

Another interesting thing is to know which is the area of the counter and how much bigger it is vs the actual working electrode especially if those measurements have been used for capacitance evaluation.

I suggest to also review different misspelled words in the manuscript.

For example CMAP (compound muscle action potential) is reported several times as CAMP instead,

Si/Au and Si/Mo is reported also as Au/Si and Mo/Si, you should find a standard single way to identify the stack.

etc.,

Reviewer #1

Summary Recommendation: The authors present their work on a novel optoelectronic device for nerve stimulation. They use multiple models (species, in/ex vivo) to characterize and validate their device and demonstrate its potential for attenuating nerve injury. Materials experts may be better able to critically assess this paper since much of the paper is about materials fabrication and characterization (Figs. 2,3,5; Supplemental figs. 1 to 12, 15), not clinical medicine. However, Figure 6 does test its clinical relevance and this seems like the primary set of experiments for this paper. The preceding work does not test the use of this device on a disease/condition, but rather much of the content leading up to this was methods/device validation. Please see my detailed comments below.

Our response: We thank the reviewer for these favorable comments and valuable suggestions. We address these issues in the point-by-point response below.

- **Comment #1:** “The overall grammar, sentence structure, and flow could be improved.”

Our response: We thank the reviewer for this valuable suggestion. We have carefully checked and improved the grammar, sentence structure and flow of the manuscript, as suggested by the reviewer.

Our modification to the manuscript: We carefully checked and improved the grammar, sentence structure and flow of the manuscript.

- **Comment #2:** “Abstract Ln38-39: how does this differ from optogenetics? it’s unclear to me how stimulation is occurring. this does not appear to be electrical or genetic. this information is presented later in the manuscript but would be helpful to have earlier on.”

Our response: We thank the reviewer for the comment. The optoelectronic neural

interface utilizes the photovoltaic effects to convert light into electrical current. Under the irradiation of a red pulsed laser, the n side of the Si diode (p⁺n) injects negative charges, and as a result induces more positive charges in the nerve tissue. This process causes cell depolarization and evokes compound muscle action potentials. Optoelectronic stimulation offers the advantages of wireless operation and eliminates the need for genetic modification which is required in optogenetic techniques. We added more information of optoelectronic stimulation in the abstract as the reviewer suggested.

Our modification to the manuscript:

We updated the description in the abstract on page 3:

“The optoelectronic neural interface utilizes photovoltaic effects to convert light into electrical current, which can induce charge redistribution and enable nerve stimulation. This innovative technique holds significant promise as a non-genetic and remote approach to modulate neural activity.”

- ***Comment #3:*** “Abstract Ln 44: stimulation for what purpose? pain? motor function? other? it sounds like this is 'for' regeneration of peripheral nerves... not 'and' regeneration of peripheral nerve”

Our response: We thank the reviewer for the comment. We apologize for the confusion. In the current work, nerve stimulation is employed for both modulating neural activity and promoting the regeneration of peripheral nerves.

Our modification to the manuscript:

We updated the abstract on page 3:

“Here, we propose a biodegradable, flexible and miniaturized silicon-based neural

interface that accomplishes transdermal optoelectronic stimulation for the modulation of neural activity and the regeneration of injured peripheral nerves.”

- *Comment #4: “Pg4Ln61: new generation devices for spinal cord and peripheral nerve stimulation are much more miniaturized now.”*

Our response: We thank the reviewer for the comment. We agree that the progress in miniaturization has greatly advanced the new generation of electrical stimulators. We intended to highlight that conventional electrical stimulators have relatively large sizes and commonly necessitate connections to bulky batteries or external equipment for power supply, which can potentially lead to undesired inflammations. Recent endeavors in device miniaturization and integration with wireless circuits based on radio frequency (RF) and inductive coupling have shown promise in addressing these concerns. We rephrased the sentences to avoid confusion.

Our modification to the manuscript:

We updated the description in page 4:

“Conventional electrical stimulators have relatively large sizes and commonly necessitate connections to bulky batteries or external equipment for power supply, which can potentially lead to undesired inflammations²¹. Recent endeavors in device miniaturization and integration with wireless circuits based on radio frequency (RF) and inductive coupling have shown promise in addressing these concerns.^{15,22,23”}

- *Comment #5: “Pg4Ln62: which drug-based methods suffer from imprecise dosing? intrathecal pumps for drug delivery are quite precise.”*

Our response: We thank the reviewer for the comment. We intended to state that drug-based methods generally have limited temporal resolution in comparison to electrical stimulations. We have revised the sentences to ensure clarity.

Our modification to the manuscript:

We updated the description in page 4:

“Moreover, drug-based methods typically exhibit limited temporal resolution compared to electrical stimulations²⁴. To further achieve miniaturized and noninvasive or minimally invasive systems, other techniques, such as ultrasound²⁵⁻²⁸, magnetoelectronics²⁹, and magnetism³⁰, have also been explored..”

- *Comment #6: “Pg5Ln75-77: include some detail about this mechanism in the opening abstract (when you contrast it to optogenetics)”*

Our response: We thank the reviewer for the comment. We have updated the description in the opening abstract.

Our modification to the manuscript:

We added description in page 3:

“The optoelectronic neural interface utilizes photovoltaic effects to convert light into electrical current, which can induce charge redistribution and enable nerve stimulation.”

- *Comment #7: “Pg5Ln81: please detail examples of when device removal would be necessary and make a biodegradable system advantageous... there is often no need to remove currently available implantable neuromodulation devices”*

Our response: We thank the reviewer for the comment. For short-term electrical stimulation (typically ranging from a few days to a few weeks) targeted at temporary treatments like tissue regeneration and pain management, the use of devices becomes unnecessary upon the completion of tissue repair or the mitigation of symptoms.

Innovative biodegradable systems can therefore offer the advantages of eliminating unnecessary material retention and avoiding the requirement for secondary surgery to remove the device, thereby maximizing therapeutic outcomes. Recent demonstrated example includes biodegradable electrical stimulators for sciatic nerve regeneration (Sci. Adv. 2020, 6(50): eabc6686; Nat. Med. 2018, 24(12): 1830), pain management (Sci. Adv. 2022, 8(40): eabp9169), etc. We added more discussions to highlight the advantages of biodegradable systems.

Our modification to the manuscript:

We added description in page 5–6:

“This is because in the context of short-term stimulation (typically ranging from a few days to a few weeks) targeted at temporary treatments like tissue regeneration and pain management, the use of devices becomes unnecessary upon the completion of tissue repair or the mitigation of symptoms. Innovative biodegradable systems can therefore offer the advantages of eliminating unnecessary material retention and avoiding the requirement for secondary surgery to remove the device, thereby maximizing therapeutic outcomes. Recent demonstrated examples include biodegradable electrical stimulators for sciatic nerve regeneration^{15,54}, pain management⁵⁵, etc.”

- ***Comment #8:*** “Pg5Ln89: please provide additional supporting references that electrical stimulation promotes regeneration... I see you do this later in the manuscript”

Our response: We thank the reviewer for the comment. Extensive studies have shown that electrical stimulation at the early stage of nerve injuries can effectively promote tissue regrowth and functional recovery in rodents (Nat. Med. 2018, 24(12): 1830; Sci. Adv. 2020, 6(50): eabc6686; Interdiscip. Neurosurg. 2021, 24: 101117; J. Neurosci. 2000, 20(7): 2602). Electrical stimulation is believed to accelerate early

Wallerian degeneration (Glia 2022,71(3): 758), stimulate calcium activity which upregulates regeneration associated genes (RAGs) through the cyclic adenosine monophosphate (cAMP) pathway (Exp. Neurol. 2020, 332: 113397), and increase the proliferation and production of neurotrophic factors of Schwann cells (Interdiscip. Neurosurg. 2021, 24: 101117), promoting nerve regeneration. We added more discussions regarding electrical stimulation and nerve regeneration.

Our modification to the manuscript:

We updated description in page 6:

“Electrical stimulation at the early stages of tissue injuries has been proven to effectively promote nerve regeneration^{15,54,60,61}. Electrical stimulation is believed to accelerate early Wallerian degeneration⁶², stimulate calcium activity which upregulates regeneration associated genes (RAGs) through the cyclic adenosine monophosphate (cAMP) pathway⁶³, and increase the proliferation and production of neurotrophic factors of Schwann cells⁶⁰, promoting nerve regeneration. Effective transdermal optoelectronic stimulation could therefore potentially provide essential electrical cues wirelessly to facilitate nerve regrowth and functional restoration, which has been rarely explored.”

- *Comment #9: “Pg6Ln93: if the device is transdermal, it seems like it could be easily removed (vs implantable) and thus, make the advantage of biodegradability less clear”*

Our response: We thank the reviewer for the comment. In our study, we implanted the optoelectronic device at the site of injured facial nerves, enabling transdermal optoelectronic stimulation to enhance nerve regeneration. The biodegradable nature of the device facilitates its gradual absorption over time, obviating the need for additional retrieval procedures. Otherwise, removal of the device through surgical means would necessitate incisions in the skin and muscle to access the implantation sites, thereby leading to secondary damage and heightened risks of infection.

Moreover, implanted devices often undergo fibrotic encapsulation, further complicating their surgical removal. In contrast, biodegradable devices offer the unique advantage of eliminating retrieval procedures and minimizing associated infection risks.

Our modification to the manuscript:

We added discussions in page 7 to highlight the advantages of biodegradable devices: “The biodegradable nature of the device can offer the unique advantage of eliminating retrieval procedures and minimizing associated infection risks.”

- **Comment #10: “Pg16 Sciatic nerve modulation: the manuscript begins with a focus on transdermal application and its advantages, but this model and others presented involve direct nerve placement (with open surgery)”**

Our response: We thank the reviewer for the comment. In our studies, we use sciatic nerve with open surgery to first evaluate the stimulation efficacy of the optoelectronic devices. The results suggest that the presence of Mo modification enables significantly improved stimulation efficacy. Based on these interesting results, we further perform transdermal stimulation on facial nerves New Zealand rabbits, as facial nerves are close to the skin and is more accessible to tissue penetrating light. In comparison to sciatic nerves, facial nerves are located closer to the skin surface, resulting in reduced absorption and scattering of penetrating light, greatly facilitating the successful implementation of transdermal optoelectronic stimulation.

Our modification to the manuscript:

We added more discussions in page 18 and 21 to improve clarity:

“As the biodegradable Mo modification layer can significantly promote photoresponse of Si diodes, we first study the modulation of the sciatic nerve of SD rats with open surgery to evaluate the stimulation efficacy of the biodegradable

optoelectronic device, by monitoring the acute response upon light illumination (Fig. 4a).”

“In comparison to sciatic nerves, facial nerves are located closer to the skin surface, resulting in reduced absorption and scattering of penetrating light, greatly facilitating the successful implementation of transdermal optoelectronic stimulation.”

- ***Comment #11: “Pg17Ln298-300: Sentence is unclear with typos”***

Our response: We thank the reviewer for the comment. We apologize for the confusion. Here we intended to state that we observe that Si devices (without a modification layer) are unable to elicit stimulation of sciatic nerves in SD rats. However, successful stimulation of sciatic nerves with Si devices (without a modification layer) was achieved in C57BL/6 mice (Nat. Biomed. Eng. 2023 7: 486), possibly due to the lower threshold voltage for sciatic nerve stimulation in C57BL/6 mice compared with SD rats used in the current study. With improved stimulation efficacy by Mo decoration, nerve activation can be accomplished on the sciatic nerve of SD rats.

Our modification to the manuscript:

We revised the sentences in page 19 to improve clarity:

“It is noted that successful stimulation of sciatic nerves with Si devices has been achieved in C57BL/6 mice⁵³, possibly due to the lower threshold voltage for sciatic nerve stimulation in C57BL/6 mice compared with SD rats used in the current study. With improved stimulation efficacy by Mo decoration, nerve activation can be accomplished on the sciatic nerve of SD rats.”

- ***Comment #12: “Pg19 Ln 332-334: consider addressing some of the possible barriers or reasons why there has rarely been attention for facial nerve application”***

Our response: We thank the reviewer for the comment. Limited studies have explored the application of optoelectronic stimulation in the treatment of facial nerve injuries, probably due to the unavailability of biodegradable miniaturized optoelectronic devices with optimal stimulation efficacy. Moreover, the complexity arising from the multiple branches of facial nerves (Regen. Biomater. 2023, 24: 507) adds to the challenges of surgical procedures performed on animal models when compared to those involving sciatic nerves. The biodegradable neural interface proposed in the current work can therefore provide a new strategy to facilitate facial nerve regeneration.

Our modification to the manuscript:

We updated description in page 21:

“Although facial nerves are more accessible to tissue-penetrating light compared to sciatic nerves, there has been rarely attention to the use of optoelectronic stimulation in facial nerve phototherapy, probably due to the unavailability of biodegradable miniaturized optoelectronic devices with optimal stimulation efficacy^{81,82.}”

- ***Comment #13: “Pg19Ln337-339: this is likely due to transdermal vs direct application. furthermore, the sciatic nerve innervates much larger muscle groups”***

Our response: We thank the reviewer for the comment. We would like to clarify that in Supplementary Fig. 11a, we utilized identical wired electrodes inserted onto the facial nerve of New Zealand rabbits and the sciatic nerve of SD rats to perform electrical stimulation and corresponding CMAP amplitude were recorded. These data can serve a baseline for subsequent optoelectronic stimulation studies. The results suggest that, compared to the sciatic nerve of SD rats, the facial nerve of New Zealand rabbits produces weaker muscle responses at the same stimulation current intensity. The reason, as suggested by the reviewer, is that the sciatic nerve in SD rats

innervates a much larger muscle group with a greater volume (Muscle Nerve, 2010, 42(2): 192; Regen. Biomater. 2023, 24: 507).

Our modification to the manuscript:

We updated description in page 22:

“The results suggest that, compared to the sciatic nerve of SD rats, the facial nerve of New Zealand rabbits produces weaker muscle responses at the same stimulation current intensity. This is likely due to the fact that the sciatic nerve in SD rats innervates a much larger muscle group with a greater volume^{83,84}.”

- *Comment #14: “Pg21Ln367-369: consider moving this to the intro/rationale”*

Our response: We thank the reviewer for the comment. We have added the information in the introduction part.

Our modification to the manuscript:

We added discussions in page 6:

“Electrical stimulation at the early stages of tissue injuries has been proven to effectively promote nerve regeneration^{15,54,60,61}. Electrical stimulation is believed to accelerate early Wallerian degeneration⁶², stimulate calcium activity which upregulates regeneration associated genes (RAGs) through the cyclic adenosine monophosphate (cAMP) pathway⁶³, and increase the proliferation and production of neurotrophic factors of Schwann cells⁶⁰, promoting nerve regeneration. Effective transdermal optoelectronic stimulation could therefore potentially provide essential electrical cues wirelessly to facilitate nerve regrowth and functional restoration, which has been rarely explored.”

- *Comment #15: “Pg22Ln379-386: this seems like the primary set of experiments for this paper.. the preceding work does not test the use of this device on a disease/condition.. rather much of the content leading up to this has been methods/device validation”*

Our response: We thank the reviewer for the comment. In this study, our main focus is on the development of a fully biodegradable, flexible, and miniaturized optoelectronic device. To assess its efficacy, we perform nerve activation experiments on the sciatic nerve of SD rats and the facial nerve of New Zealand rabbits. We test the device on a model of injured facial nerves in New Zealand rabbits to evaluate its therapeutic capabilities. The results demonstrate that transdermal optoelectronic stimulation using the biodegradable optoelectronic device enhances functional recovery in cases of facial nerve injuries.

Our modification to the manuscript:

We updated discussions in page 7:

“Successful functional recovery of injured facial nerves is achieved by transdermal optoelectronic stimulation, to demonstrate the therapeutic efficacy of the device on facial nerve regeneration.”

- *Comment #16: “Supplementals: can the supplemental figures be condensed or combined somehow? 20 supplemental figures are a bit much since many of these figures have small amounts of data presented”*

Our response: We thank the reviewer for the valuable suggestion. We have consolidated supplementary figures pertaining to related topics, thereby decreasing the overall quantity of supplementary figures.

Our modification to the manuscript:

We have consolidated some supplementary figures pertaining to related topics:

Supplementary Fig. 6 | **a** Photocurrent shows high retention after 12000 pulse cycles (635 nm, 0.95 W/cm^2 , 10 Hz, pulse width 10 ms). I : measured photocurrent; I_0 : initial photocurrent. $n = 3$ independent experiments. **b** Light transmission through Au (~ 10 nm) or Mo (~ 10 nm) modification layers. **c** Photoresponses under illumination (635 nm, 10 Hz, pulse width 10 ms, 0.95 W/cm^2) incident from the front side and back side (transmitted through the modification layer and separated extended electrodes) of Si/Mo devices.

Supplementary Fig. 9 | **a** Representative X-ray photoelectron spectroscopy (XPS) analysis of the Mo modification layer of Si/Mo devices before CV cycling. **b, c, d** Three-dimensional reconstruction of the p⁺ side of the Si/Mo, Si/Au, and Si devices obtained by scanning with an atomic force microscope (AFM).

Supplementary Fig. 11 | **a** CMAP amplitudes of muscles evoked by stimulating the sciatic nerve in rats and the facial nerve in New Zealand rabbits with different currents. $n = 3$ independent experiments. **b** Si/Mo devices retrieved from the implantation site after 4 days of implantation. The device remains almost intact. **c** Measured photovoltage of Si/Mo devices before and after implantation over a 6-day time frame (light intensity: 0.95 W/cm^2). **d** Images of the anatomy of the facial nerve, crush injury, and device implantation.

- **Comment #17:** *“Facial nerve crush injury/S16: how is the nerve crush injury standardized for replicates? it looks like this is performed manually and would likely involve variable pressures of crush injury”*

Our response: We thank the reviewer for the comment. We use hemostats to produce crush injury to the facial nerve at the same location. The facial nerve is clamped twice using toothless hemostatic forceps for 30 seconds, with 10-second intervals between the two clamps. The hemostat’s ring handles have a ratcheted

locking mechanism that secures the jaws in a fixed position. We utilized the same position of the jaws of the hemostat to create crush injuries and employed the ratchet mechanism to lock the jaws in identical positions, ensuring consistent force and therefore reproducible degree of injuries for each experiment.

Our modification to the manuscript:

We updated the methodology session in page 36:

“The procedure involved exposing the buccal branch of the facial nerve and inducing crush injuries by clamping the nerve twice with toothless hemostatic forceps for 30 seconds, with 10-second intervals between clamps. The hemostat’s ring handles have a ratcheted locking mechanism that secures the jaws in a fixed position. We utilize the same position of the jaws of the hemostat to create crush injuries and employ the ratchet mechanism to lock the jaws in identical positions, ensuring consistent force and therefore reproducible degree of injuries for each experiment.”

Reviewer #2

Summary Recommendation: The present manuscript presents a flexible neural interface activated by optoelectronic transdermal stimulation. The proposed use is for electroneurostimulation following peripheral axotomy, as a method to enhance recovery of motor function after crush of the facial nerve in rabbits.

Overall, the work is well presented and organized, so that the construction of the implantable device is innovative and effective in eliciting compound muscle action potentials in a reproducible fashion.:

Our response: We thank the reviewer for these favorable comments and valuable suggestions.

- **Comment #1:** “Nevertheless, the authors did not take into account the Wallerian degeneration process that takes place distally to the site of axotomy, in the present case, the site of the nerve crush. Since the severed axons degenerate post-injury, what is the role of electrostimulation?”

Our response: We thank the reviewer for the comment. Previous studies reported that electrical stimulation at the early stage of nerve injury (1 hour to a few days) can significantly promote regeneration at the later stage (J. Neurosci. 2000, 20(7): 2602; Exp. Neurol. 2020, 323: 113074; Nat. Med. 2018, 24(12): 1830). Although the mechanisms are not fully understood, the positive effects of electrostimulation on Wallerian degeneration and nerve regeneration processes have been suggested. For example, electrical stimulation in the early recovery phase have been reported to expedite Wallerian degeneration (Glia 2022, 71(3): 758). The accelerated degeneration and clearance of axons and myelin sheath can assist creating a more favorable microenvironment for nerve regeneration. Moreover, electrical stimulation at the early stage could offer prompt and continuous upregulation of regeneration associated genes (Exp. Neurol. 2020, 323:113074; Interdiscip. Neurosurg. 2021, 24:

101117). These effects are expected expedite staggered regeneration of axons across the injury site and therefore facilitates more rapid axonal regrowth.

We perform an assessment of the degree of degeneration of myelinated axons 7 days postimplantation of the control and Si/Mo OS-4d groups. Toluidine blue staining and TEM analysis of the transversely sectioned nerve segments, located 10 mm distal from the crush site on day 7 postoperatively are given in Supplementary Fig. 16. In the control group, a substantial amount of degenerating myelinated axons (marked by the yellow arrow) and some intact myelinated axons (marked by the red arrow) are observed. In contrast, in the Si/Mo OS-4d group, axonal degeneration is more advanced which is evidenced by the greater clearance of myelin and axons, resulting in significantly fewer degenerating myelinated nerve fibers and few intact axons. The statistical results show significantly less density of myelinated nerve fibers (including both degenerating and intact axons) in the Si/Mo OS-4d group compared with the control group ($p < 0.01$), based on the toluidine blue staining images. The results are consistent with the previous report (Glia 2022, 71(3): 758), suggesting a 4-day optoelectronic stimulation at the early phase could accelerate Wallerian degeneration, and thereby creating a more favorable microenvironment for nerve regrowth.

Our modification to the manuscript:

We added Supplementary Fig. 16 and update the discussions on page 27–28:

Supplementary Fig. 16 | Effects of optoelectronic stimulation on Wallerian degeneration in facial nerve crush injury. **a** Toluidine Blue staining of the transversely sectioned nerve segment located 10 mm distal from the crush site on day 7 postoperatively. **b** TEM images of the transversely sectioned nerve segments located 10 mm distal from the crush site on day 7 postoperatively. **c** The density of myelinated nerve fibers based on toluidine blue staining results. Myelinated nerve fibers include both the degenerating (indicated by the yellow arrow) and intact (indicated by the red arrow) nerve fibers on day 7 postoperatively. Groups: Control (no treatment), Si/Mo OS-4d (optoelectronic stimulation for 4 days with Si/Mo devices). $n = 3$ independent experiments. All data are presented as mean \pm s.e.m.

“The enhancement of long-term axonal growth through electrical stimulation in the early recovery phase may be attributed to the expedited Wallerian degeneration⁶² and the prompt and continuous upregulation of regeneration associated genes^{60,89}, which accelerates staggered regeneration of axons across the injury site and therefore facilitates more rapid axonal regrowth. We perform an assessment of the degree of degeneration of myelinated axons 7 days postimplantation of the control and Si/Mo OS-4d groups. Toluidine blue staining and TEM analysis of the transversely sectioned nerve segments, located 10 mm distal from the crush site on day 7 postoperatively are given in Supplementary Fig. 16. In the control group, a substantial amount of degenerating myelinated axons (marked by the yellow arrow) and some intact myelinated axons (marked by the red arrow) are observed. In contrast, in the Si/Mo OS-4d group, axonal degeneration is more advanced which is evidenced by the greater clearance of myelin and axons, resulting in significantly fewer degenerating myelinated nerve fibers and few intact axons. The statistical results show significantly less density of myelinated nerve fibers (including both degenerating and intact axons) in the Si/Mo OS-4d group compared with the control group ($p < 0.01$), based on the toluidine blue staining images. The results are consistent with the previous report⁶² suggesting a 4-day optoelectronic stimulation at the early phase could accelerate Wallerian degeneration, and thereby creating a more favorable microenvironment for nerve regrowth.”

- ***Comment #2*** *“In turn, a four-day therapy is highly controversial in terms of yielding long-term axonal growth acceleration.”*

Our response: We thank the reviewer for the comment. It has been reported that electrical stimulation at the early stage of nerve injury (1 hour to a few days) can significantly promotes regeneration at the later stage. For example, electrical stimulation for 1 h right after nerve injury can significantly promote nerve regeneration in crush and transection models of nerve injury after 6–10 weeks (J. Neurosci. 2000, 20(7): 2602; Exp. Neurol. 2020, 323: 113074); the clinical study of

1-h electrical stimulation immediately after surgery has also been shown to greatly enhance the recovery of sensory function in human hands by 5-6 months (Ann. Neurol. 2015, 77(6): 996); moreover, the application of electrical stimulation for 1 h each day over a period of six days in the early phase of recovery significantly improved the speed and extent of sciatic nerve regeneration after 10 weeks (Nat. Med. 2018, 24(12): 1830). The enhancement of long-term axonal growth through electrical stimulation in the early recovery phase may be attributed to the expedited Wallerian degeneration (Glia 2022, 71(3): 758) and the prompt and continuous upregulation of regeneration associated genes (Exp. Neurol. 2020, 323:113074; Interdiscip. Neurosurg. 2021, 24: 101117), which expedites staggered regeneration of axons across the injury site and therefore facilitates more rapid axonal regrowth.

Our modification to the manuscript:

We added the discussions on page 27:

“The enhancement of long-term axonal growth through electrical stimulation in the early recovery phase may be attributed to the expedited Wallerian degeneration⁶² and the prompt and continuous upregulation of regeneration associated genes^{60,89}, which accelerates staggered regeneration of axons across the injury site and therefore facilitates more rapid axonal regrowth.”

- **Comment #3:** “The immunohistochemical analysis of the nerves, shown in Figure 6, Figure S17, and Figure S19 are of below-average quality. There is a need to indicate the site of injury and the proximal and distal stumps.”

Our response: We thank the reviewer for the valuable suggestions. We reperformed the immunohistochemical staining of the nerve segments and improved the quality of the data. We also labelled the site of injury, proximal and distal stumps of the nerve segments.

Our modification to the manuscript:

We updated Fig. 6a, b, c, and Supplementary Fig. 12:

Fig. 6 | Evaluations of regenerated nerve segments at 4 and 8 weeks after phototherapy with the biodegradable optoelectronic neural interfaces. a, b Immunofluorescence staining images of longitudinally sectioned regenerated nerve segments located 5 mm distal from the crush sites at 4 weeks postoperatively. Phototherapy is applied 1-h daily for 1 or 4 days (635 nm, 10 Hz, pulse width 10 ms, 0.95 W/cm²). Groups: Control (no treatment), PS-4d (photostimulation for 4 days), Si OS-4d (optoelectronic stimulation for 4 days with Si devices), Si/Mo OS-1d (optoelectronic stimulation for 1 day with Si/Mo devices), Si/Mo OS-4d (optoelectronic stimulation for 4 days with Si/Mo devices). **a** Immunohistochemical staining: axons (NF200, green), myelin sheaths (MBP, red), and nuclei (DAPI, blue). **b** Immunohistochemical staining: newly grown axons (GAP-43, green), and nuclei (DAPI, blue). **c** Immunofluorescence staining images of transversely sectioned regenerated nerve segments located 10mm distal from the crush sites at 8 weeks postoperatively. Immunohistochemical staining: axons (NF200, green), myelin sheaths (MBP, red), and nuclei (DAPI, blue).

Supplementary Fig. 12 | Immunofluorescence staining images of the longitudinally sectioned regenerated nerve segments at 4 weeks postoperatively.

a Immunohistochemical staining: axons (NF200, green), myelin sheaths (MBP, red), and nuclei (DAPI, blue). **b** Immunohistochemical staining: newly grown axons (GAP-43, green), and nuclei (DAPI, blue). White arrow indicates the site of crush injuries. Phototherapy is applied 1-h daily for 1 or 4 days. Groups: Control (no treatment), PS-4d (photostimulation for 4 days), Si OS-4d (optoelectronic stimulation for 4 days with Si devices), Si/Mo OS-1d (optoelectronic stimulation for 1 day with Si/Mo devices), Si/Mo OS-4d (optoelectronic stimulation for 4 days with Si/Mo devices). n = 4 independent experiments.

- ***Comment #4:*** “The increase of S100 immunolabeling appears to be linked to fibrosis and not with an enhancement of function/recovery. There is a need for a time-course evaluation of the morphological recovery, including markers of axonal regeneration, such as CGRP or GAP-43. Staining for MBP could help evaluate myelination.”

Our response: We thank the reviewer for this valuable suggestion. We reperformed the immunohistochemical staining of the nerve segments as suggested by the reviewer, with GAP-43, MBP and NF200 for the early stage (4 weeks postimplantation, as GAP-43 is expected to be highly expressed in newly formed axons), and MBP and NF200 for the later stage (8 weeks postimplantation), and the results are updated in Fig. 6a, b, c and Supplementary Fig. 12. The immunofluorescence staining images of the longitudinally sectioned nerve segments located 5 mm distal from the crush sites at 4 weeks postoperatively suggest that, greater number of directional axons denoted by both GAP-43 and NF200 (green fluorescence) is observed in the Si/Mo OS-4d group compared with the other groups (Fig. 6a, b and Supplementary Fig. 12). Moreover, the density of myelin sheaths denoted by MBP (red fluorescence) surrounding the nerve fibers (NF200, green fluorescence) is also greater in the Si/Mo OS-4d group compared to the other groups (Fig. 6a and Supplementary Fig. 12a). Furthermore, the immunofluorescence staining images of transversely sectioned nerve segments located 10mm distal from the crush sites at 8 weeks postoperatively (Fig. 6 c) indicate that, myelination in the Si/Mo OS-4d and Si/Mo OS-1d groups is more mature than the other groups, in the sense that

the presence of fluorescent signals of myelin sheaths (MBP, red fluorescence) are greater surrounding the nerve fascicles regions, indicating a more profound and expedited process of myelination. These results provide evidence of the beneficial effects of the biodegradable optoelectronic neural interface for facial nerve regeneration.

Our modification to the manuscript:

We updated Fig. 6a, b, c and Supplementary Fig. 12 and discussions in page 27:

Fig. 6 | Evaluations of regenerated nerve segments at 4 and 8 weeks after phototherapy with the biodegradable optoelectronic neural interfaces. a, b Immunofluorescence staining images of longitudinally sectioned regenerated nerve segments located 5 mm distal from the crush sites at 4 weeks postoperatively. Phototherapy is applied 1-h daily for 1 or 4 days (635 nm, 10 Hz, pulse width 10 ms, 0.95 W/cm²). Groups: Control (no treatment), PS-4d (photostimulation for 4 days), Si OS-4d (optoelectronic stimulation for 4 days with Si devices), Si/Mo OS-1d (optoelectronic stimulation for 1 day with Si/Mo devices), Si/Mo OS-4d (optoelectronic stimulation for 4 days with Si/Mo devices). **a** Immunohistochemical staining: axons (NF200, green), myelin sheaths (MBP, red), and nuclei (DAPI, blue). **b** Immunohistochemical staining: newly grown axons (GAP-43, green), and nuclei (DAPI, blue). **c** Immunofluorescence staining images of transversely sectioned regenerated nerve segments located 10mm distal from the crush sites at 8 weeks

postoperatively. Immunohistochemical staining: axons (NF200, green), myelin sheaths (MBP, red), and nuclei (DAPI, blue).

Supplementary Fig. 12 | Immunofluorescence staining images of the longitudinally sectioned regenerated nerve segments at 4 weeks postoperatively.

a Immunohistochemical staining: axons (NF200, green), myelin sheaths (MBP, red), and nuclei (DAPI, blue). **b** Immunohistochemical staining: newly grown axons (GAP-43, green), and nuclei (DAPI, blue). White arrow indicates the site of crush injuries. Phototherapy is applied 1-h daily for 1 or 4 days. Groups: Control (no treatment), PS-4d (photostimulation for 4 days), Si OS-4d (optoelectronic stimulation for 4 days with Si devices), Si/Mo OS-1d (optoelectronic stimulation for 1 day with Si/Mo devices), Si/Mo OS-4d (optoelectronic stimulation for 4 days with Si/Mo devices). n = 4 independent experiments.

We added discussions in page 25:

“The immunofluorescence staining images of the longitudinally sectioned nerve segments located 5 mm distal from the crush sites at 4 weeks postoperatively suggest that, greater number of directional axons denoted by both GAP-43 and NF200 (green fluorescence) is observed in the Si/Mo OS-4d group compared with the other groups (Fig. 6a, b and Supplementary Fig. 12). Moreover, the density of myelin sheaths denoted by MBP (red fluorescence) surrounding the nerve fibers (NF200, green fluorescence) is also greater in the Si/Mo OS-4d group compared to the other groups (Fig. 6a and Supplementary Fig. 12a). Furthermore, the immunofluorescence staining images of transversely sectioned nerve segments located 10mm distal from the crush sites at 8 weeks postoperatively (Fig. 6 c) indicate that, myelination in the Si/Mo OS-4d and Si/Mo OS-1d groups is more mature than the other groups, in the sense that the presence of fluorescent signals of myelin sheaths (MBP, red fluorescence) are greater surrounding the nerve fascicles regions, indicating a more profound and expedited process of myelination. These results provide evidence of the beneficial effects of the biodegradable optoelectronic neural interface for facial nerve regeneration.”

- ***Comment #5:*** “The CMAP data connected with the immunolabeled material is extremely variable, showing strange groups of data within each experimental group

(p. ex., Figure 6C). Infinite latency in control nerves is not acceptable.

Our response: We thank the reviewer for the comment. We speculate that the large variation in the CMAP results observed at 4 weeks postimplantation is probably due to the fact that this timeframe falls within the early stage of nerve regeneration, where re-innervation to the target muscles is still ongoing. Instances of newly established re-innervation can result in significantly greater measured CMAP values compared to those where re-innervation has not yet occurred, resulting in large variation of the recorded data. Such variation in CMAP measurement has been reported before in rodents especially at the initial phase of nerve regrowth (e.g., ACS Appl. Mater. Interfaces, 2020, 12 (33): 36860; Adv. Mater. 2024, 36: 2305374). Furthermore, more successful cases of re-innervation, denoted by greater CMAP amplitudes, are evident in the Si/Mo OS-4d and Si/Mo OS-1d groups (Supplementary Fig. 15), indicating the beneficial effects of optoelectronic stimulation. In contrast, in the control group, the probability of successfully collecting measurable CMAP is lower compared to the other groups and these groups were incorporated as having latency of infinity. We agree with the reviewer that these groups should not be included for statistical analysis. By 8 weeks post-implantation, the variation in CMAP measurement significantly decreases, particularly in the Si/Mo OS-4d group, suggesting that nerve regeneration has progressed to a more advanced stage (Fig. 6e).

We re-evaluated the CMAP data at 4 and 8 weeks postimplantation, and updated the data in Fig. 6d, e, f and Supplementary Fig. 15. The CMAP data at 4 weeks postoperatively demonstrate relatively large variation, probably due to incomplete tissue regeneration and re-innervation to the target muscles at such an early stage. Nevertheless, more instances of greater CMAP amplitudes are observed in the Si/Mo OS-4d and Si/Mo OS-1d groups, suggesting a greater chance of faster nerve re-innervation. Statistically, the measured CMAP amplitude (peak-peak value) at 4 weeks postoperatively is significantly greater in the Si/Mo OS-4d and Si/Mo OS-1d groups ($p < 0.01$ or $p < 0.05$) as compared to the other groups (Supplementary Fig. 15a). The latency of the Si/Mo OS-4d and Si/Mo OS-1d groups are lower compared

to the Si OS-4d group ($p < 0.001$), and the latency of the Si/Mo OS-1d group is lower than that of the Control and PS-4d groups ($p < 0.05$) (Supplementary Fig. 15b).

At 8 weeks postimplantation when nerve regeneration has progressed to a more mature stage, the Si/Mo OS-4d group exhibits significantly greater amplitude compared to the control, PS-4d, Si OS-4d and Si/Mo OS-1d groups ($p < 0.001$) (Fig. 6e), and the amplitude is approaching that of the normal side (wild-type, WT). The Si/Mo OS-1d group also demonstrates significantly greater CMAP amplitude compared with the control ($p < 0.01$) and Si OS-4d groups ($p < 0.05$). No significant difference in latency is observed among the Si/Mo OS-4d and Si/Mo OS-1d groups compared to the WT, control and Si OS-4d groups (Fig. 6f). Collectively, the CMAP and immunohistochemical results suggest that the optoelectronic device with Mo decoration can facilitate functional recovery by providing electrical cues at the initial phase of nerve injuries.

Our modification to the manuscript:

We updated Fig. 6d, e, f and Supplementary Fig. 15:

Fig. 6 | **d** Representative evoked CMAPs at 8 weeks postoperatively. Red triangle indicates the point of light stimulation. **e** CMAP amplitude at 8 weeks postoperatively. **f** Latency of CMAPs at 8 weeks postoperatively.

Supplementary Fig. 15 | Statistical analysis of CMAP at 4 weeks postoperatively.

a CMAP amplitude at 4 weeks postoperatively. **b** Latency of CMAPs at 4 weeks postoperatively. Phototherapy is applied 1-h daily for 1 or 4 days. Groups: Control (no treatment), PS-4d (photostimulation for 4 days), Si OS-4d (optoelectronic stimulation for 4 days with Si devices), Si/Mo OS-1d (optoelectronic stimulation for 1 day with Si/Mo devices), Si/Mo OS-4d (optoelectronic stimulation for 4 days with Si/Mo devices). $n = 4$ independent experiments.

We updated discussions in page 26–27:

“The CMAP data at 4 weeks postoperatively demonstrate relatively large variation, probably due to incomplete tissue regeneration and re-innervation to the target muscles at such an early stage. Nevertheless, more instances of greater CMAP amplitudes are observed in the Si/Mo OS-4d and Si/Mo OS-1d groups, suggesting a greater chance of faster nerve re-innervation. Statistically, the measured CMAP amplitude (peak-peak value) at 4 weeks postoperatively is significantly greater in the Si/Mo OS-4d and Si/Mo OS-1d groups ($p < 0.01$ or $p < 0.05$) as compared to the other groups (Supplementary Fig. 15a). The latency of the Si/Mo OS-4d and Si/Mo OS-1d groups are lower compared to the Si OS-4d group ($p < 0.001$), and the latency of the Si/Mo OS-1d group is lower than that of the Control and PS-4d groups ($p < 0.05$) (Supplementary Fig. 15b).

At 8 weeks postimplantation when nerve regeneration has progressed to a more mature stage, the Si/Mo OS-4d group exhibits significantly greater amplitude compared to the control, PS-4d, Si OS-4d and Si/Mo OS-1d groups ($p < 0.001$) (Fig. 6e), and the amplitude is approaching that of the normal side (wild-type, WT). The Si/Mo OS-1d group also demonstrates significantly greater CMAP amplitude compared with the control ($p < 0.01$) and Si OS-4d groups ($p < 0.05$). No significant difference in latency is observed among the Si/Mo OS-4d and Si/Mo OS-1d groups compared to the WT, control and Si OS-4d groups. Collectively, the CMAP and immunohistochemical results suggest that the optoelectronic device with Mo decoration can facilitate functional recovery by providing electrical cues at the initial phase of nerve injuries.”

Reviewer #3

Summary Recommendation: Dear P. Sun and co-authors, I have reviewed your paper concerning the development of a flexible transdermal optoelectronic device aiming to the modulation and regeneration of peripheral nerve. The overall study sounds appealing, original and tends to strengthen the hardcore research made nowadays to find a straightforward viable method to treat these diseases. The wireless method sounds intelligent and elegant indeed if compared to other sources of relatively minimally invasive techniques (RF, temporal interference, ultrasound, etc.) due to spatial resolution and other principle issues.

The handful of work shows also the importance in assessing heat transfer phenomena during the evaluation of the mechanism of the device to decouple thermal phenomena from capacitive ones.

I kindly suggest this paper to be published in Nature Communications eventhough there are different points, in my opinion which must be definitely clarified and deeper discussed before the publication.

Our response: We thank the reviewer for these favorable comments. We address these issues in the point-by-point response below.

- *Comment #1: “I would like you to compare the light intensity used from you with actual quantitative numbers of light safety limits after which the illumination is considered hazardous.”*

Our response: We thank the reviewer for the valuable suggestion. The peak light intensity and average light intensity of the red pulsed laser (635 nm, 10 Hz, 10 ms) used in current animal experiments are 0.95 W/cm² and 0.19 W/cm² respectively. According to the IEC 60825-1:2014 international standard on safety of laser products, the maximum permissible exposure of the skin to laser radiation is 0.2 W/cm², in the wavelength of 400-700 nm, with the exposure time of 10³ to 3×10⁴ s (IEC 60825-1:2014 international standard: 66). The average light intensity (0.19 W/cm²) used in the current study is within the limit. Furthermore, pulsed lasers are considered safer

than continuous wave lasers because they can minimize thermal effects at the same average intensity (Lasers Surg. Med. 2010, 42(6): 450). Our results also show no significant temperature rise after one hour of skin exposure to radiation, suggesting that the light intensity employed in the current studies has minimal adverse effects.

Our modification to the manuscript:

We added the discussions on page 24:

“The average power density of the pulsed laser (0.19 W/cm²) employed in the current studies falls within the skin exposure limit (0.2 W/cm²)⁸⁸, suggesting minimal adverse effects.”

- **Comment #2:** “The Si/Mo modified device, seems to be so far the best solution to your scientific aim but have you performed deeper experiment to assess which are the electrochemical reactions which lead to a better performance of the device? From cyclic voltammetry is clear that Mo oxidizes in mimicking physiological solutions. What about possible oxygen reduction which will lead to reactive oxygen species production? Have you evaluated this possibility?”

Our response: We thank the reviewer for the valuable suggestion. We speculate that the potential oxidation of the Mo modification layer could potentially allow more effective charge injection at the cathodic side through associated electrochemical reactions or photocapacitive effects, enabling improved charge injection efficiency and stimulation efficacy. Oxygen reduction to reactive oxygen species could be one possible electrochemical reaction at the cathodic site, as suggested from previous studies (Nat. Mater. 2022, 21: 647; Sci. Adv. 2023, 9: eabq7750). We therefore performed ROS detection upon photo irradiation for 10 mins on the surface of the Si, Si/Mo and Si/Au devices, but no significant difference in ROS concentration was detected. We speculate that the amount of potentially generated ROS could be too small to be detected in the current case.

ROS detection: photo irradiation for 10 mins, the solution on the surface of the Si, Si/Mo and Si/Au devices are sampled by DCFH-DA (100 T, Solarbio) assay.

Our modification to the manuscript:

We updated the discussions on page 17:

“Based on these results, we speculate that Mo decoration layer at the p⁺ side can potentially allow stronger electrochemical reactions at the device/solution interface and induce pseudocapacitive behavior, which is beneficial for efficient charge injection by improving both faradaic and capacitive contributions. Considering charge conservation, corresponding charge injection at the n side is expected to increase as well, promoting charge injection and thereby stimulation efficacy, through photocapacitive effects and/or associated electrochemical reactions. For example, reactive oxygen species (ROS) could be produced at the interface that modulates neural activity^{52,77}. Nevertheless, the amount of potentially generated ROS could be very small and was not detected in the current experiments.”

- **Comment #3:** *“In Figure S6, P. Sun and coauthors show a comparison between Si/Mo and Si/Au (10nm of decoration) regarding high retention index and I would say that the only reason for which Au fails more is due to highly likely recrystallization of*

the layer in Cl⁻ containing solutions which leads to faster degradation of the material”

Our response: We thank the reviewer for the valuable suggestion. It has been observed that Cl⁻ ions can be adsorbed onto the surface of Au thin films in chloride-containing solutions (J. Electrochem. Soc. 2005, 152 (6): D103), which could influence the microstructure of Au thin films, as the reviewer suggested. Furthermore, given that Au thin films have shown to adhere poorly to the Si surface (Int. J. Solids Struct. 2019, 180–181: 30; PNAS 2010, 107 (22): 9950), we speculate that potential microscale delamination under optoelectronic stimulation in chloride solutions could also contribute to faster degradation.

Our modification to the manuscript:

We added the discussion on page 14:

“This may arise from the adsorption of chloride ions on the surface of Au films⁷¹, which could influence their microstructure. Additionally, potential microscale delamination could also contribute, as Au thin films are known to exhibit poor adhesion to Si surfaces^{72,73}.”

- ***Comment #4: “In Figure S8 would be better to compare all the data in terms of current density and not just current because readers will be interested in the area of your samples due to couple of order of magnitude difference in current. And also please specify the Potential vs what (i.e. E(V) vs Ag/AgCl since this is the second species electrode you have used as reference)”***

Our response: We thank the reviewer for the valuable suggestion. The samples for tests were all immersed in solutions with an area of 1 cm² and the reference electrode used was Ag/AgCl. We have updated the information in Fig. 3 and Supplementary Fig. 7 by reporting data in terms of current density and specifying potential vs. Ag/AgCl.

Our modification to the manuscript:

We updated Fig. 3a:

Fig. 3 | a Representative cyclic voltammetry (CV) curves of Si/Mo, Si/Au, and Si devices.

We updated Supplementary Fig. 7:

Supplementary Fig. 7 | **a** Schematic diagram of the electrochemical measurement. **b**, **c** and **d** Representative CV curves of Si/Mo devices, Si/Au devices and Si devices (no modification layer).

- **Comment #5:** *“In Figure S9 would be better to report if possible the Bode plot and not only the Nyquist in order to give an image of the phase as well. And please add the phase in Figure 3 of the actual manuscript.”*

Our response: We thank the reviewer for the valuable suggestion. We have modified Fig. 3 and Supplementary Fig. 8 (used to be Supplementary Fig. 9) by adding phase diagrams and bode diagrams respectively.

Our modification to the manuscript:

We have modified Fig. 3 and Supplementary Fig. 8:

Fig. 3 | **Characteristics of interface modification layers of Si-based optoelectronic**

devices. **a** Representative cyclic voltammetry (CV) curves of Si/Mo, Si/Au, and Si devices. **b** Representative Bode plots of electrochemical impedance (EIS) at the p⁺ side of Si/Mo, Si/Au, and Si devices. **c** Representative X-ray photoelectron spectroscopy (XPS) analysis results of the Mo modification layer surface of Si/Mo devices after 10 cycles of CV (-0.4 V–0.4 V). **d** Surface roughness of the modification layers (Mo or Au) or the p⁺ side of the Si surface (no modification). n = 3 in each group. All data are presented as mean ± s.e.m.

Supplementary Fig. 8 | Equivalent circuit models of electrochemical impedance.

a Equivalent circuit of the electrochemical impedance measurement. The parallel

circuit of R_1 and CPE_1 (constant phase element) is equivalent to the interface between Si and the metallic modification film. The parallel circuit of R_2 (charge transfer resistance) and CPE_2 is equivalent to the interface between the metallic modification film and PBS. R_3 is solution resistance. **b** Circuit model fitting parameters. **c** Measured electrochemical impedance and model fitting results (solid lines) of Si/Mo and Si/Au devices.

- ***Comment #6:*** *“In Figure S10 I would question about the XPS of qualitative data on a 10nm thick layer because what the authors are looking at could even be sputtered surface during the XPS analysis itself.”*

Our response: We thank the reviewer for the comment. The XPS technique is widely adopted for the surface analysis of metallic thin films with thickness of a few tens of nanometers (e.g., *Materialia* 2022, 24: 101511; *Adv. Funct. Mater.* 2014, 24: 645; *J. Korean Phys. Soc.* 2011, 58(3): 509). As depth profiling was not conducted in the current experiments, ion-beams were not used for etching the materials, and sputtering of the sample surface is therefore expected to be minimal. Moreover, samples were placed in the high vacuum system ($< 10^{-6}$ Pa), oxide formation during the measurement is probably unlikely. The XPS results indicate that oxides are present at the deposited Mo thin films, which is consistent with previous studies (*Materialia* 2022, 24: 101511; *Adv. Funct. Mater.* 2014, 24: 645).

Our modification to the manuscript:

We added the description on page 17:

“X-ray photoelectron spectroscopy (XPS) analysis is widely adopted for surface analysis of metallic thin films^{75,76}. The XPS measurement of the surface of the Mo modification layer demonstrates slightly increased amounts of Mo^{4+} and Mo^{6+} after CV cycling (Fig. 3c and Supplementary Fig. 9a), suggesting the presence of potential oxidation.”

- ***Comment #7: “Another interesting thing is to know which is the area of the counter and how much bigger it is vs the actual working electrode especially if those measurements have been used for capacitance evaluation.”***

Our response: We thank the reviewer for the comment. We used a Pt mesh counter electrode with an area of 1 cm². The area of working electrodes used in the CV and EIS tests were also 1 cm². The mesh structure of the Pt counter is expected to have similar effective surface area to the working electrodes.

Our modification to the manuscript:

We added the description on page 33:

“PBS served as the electrolyte, the Si diode (p⁺n, on SOI) with or without modification layers served as the working electrode, Ag/AgCl was used as the reference electrode, and the mesh Pt electrode (with an area of 1 cm²) was used as the counter electrode.”

- ***Comment #8: “I suggest to also review different misspelled words in the manuscript. For example CMAP (compound muscle action potential) is reported several times as CAMP instead, Si/Au and Si/Mo is reported also as Au/Si and Mo/Si, you should find a standard single way to identify the stack.”***

Our response: We thank the reviewer for the valuable suggestion. We have carefully checked the spelling of the manuscript and corrected the inconsistency of CMAP, Si/Au, and Si/Mo.

Our modification to the manuscript:

We have carefully checked the spelling of the manuscript and corrected the inconsistency of CMAP, Si/Au, and Si/Mo.

REVIEWERS' COMMENTS

Reviewer #2 (Remarks to the Author):

The revised version of the manuscript is remarkably improved, and this reviewer's suggestions have been fully and appropriately addressed. I would suggest that the authors revise lines 117-122 and 336-337 as there are repetitive statements/phrases. Otherwise, I consider the current version of the paper to be a real advance in the field and hope that the device can be further tested in primates with a view to future use in patients.

Reviewer #3 (Remarks to the Author):

I have reviewed the resubmitted manuscript and all the comments/concerns I have brought up have been carefully addressed. I suggest the acceptance of the manuscript with no further change.

Reviewer #2

Summary Recommendation: The revised version of the manuscript is remarkably improved, and this reviewer's suggestions have been fully and appropriately addressed. I would suggest that the authors revise lines 117-122 and 336-337 as there are repetitive statements/phrases. Otherwise, I consider the current version of the paper to be a real advance in the field and hope that the device can be further tested in primates with a view to future use in patients.

Our response: We thank the reviewer for these favorable comments and valuable suggestions. We have carefully verified the sentences and modified the repetitive statements.

Our modification to the manuscript:

We updated the description on page 7:

“Nerve activation is achieved by irradiating tissue-penetrating red light (635 nm) on the sciatic nerve of Sprague-Dawley (SD) rats and the facial nerve of New Zealand rabbit (Fig. 1). Successful functional recovery of injured facial nerves is achieved by transdermal optoelectronic stimulation. These results offer new avenues for the development of biodegradable, efficient and miniaturized optoelectronic devices for transdermal neural modulation and associated regenerative medicine.”

And we deleted repetitive statement on page 19:

“With improved stimulation efficacy by Mo decoration, nerve activation can be accomplished on the sciatic nerve of SD rats with a greater stimulation threshold.”